# Training Language Models to Reason Efficiently

**Daman Arora**
Carnegie Mellon University
damana@andrew.cmu.edu

**Andrea Zanette**
Carnegie Mellon University
zanette@cmu.edu

## Abstract

Scaling model size and training data has led to great advances in the performance of Large Language Models (LLMs). However, the diminishing returns of this approach necessitate alternative methods to improve model capabilities, particularly in tasks requiring advanced reasoning. Large reasoning models, which leverage long chain-of-thoughts, bring unprecedented breakthroughs in problem-solving capabilities but at a substantial deployment cost associated to longer generations. Reducing inference costs is crucial for the economic feasibility, user experience, and environmental sustainability of these models.

In this work, we propose to train large reasoning models to reason efficiently. Our method incentivizes models to minimize unnecessary computational overhead while largely maintaining accuracy, thereby achieving substantial deployment efficiency gains. It enables the derivation of a family of reasoning models with varying efficiency levels, controlled via a single hyperparameter. Experiments on two open-weight large reasoning models demonstrate significant reductions in inference cost while preserving most of the accuracy.

## 1 Introduction

Large language models (LLMs) have made significant advancements by pre-training larger models with extensive datasets (Kaplan et al., 2020), but this approach faces diminishing returns due to limited high-quality training data. An alternative to improve model capabilities, especially in domains involving careful reasoning, involves allowing models to "think" before answering, as seen in frontier reasoning models like OpenAI's o1 (OpenAI et al., 2024), Gemini 2.0 Flash Thinking Experimental, and DeepSeek-R1 (Guo et al., 2025). These models produce intermediate tokens during inference, collectively referred to as *chain-of-thoughts* (Wei et al., 2022), to perform additional computations before returning an answer. The process of generating a long chain of thought before answering the user query is called *reasoning*. More precisely, *large reasoning models* with chain-of-thoughts capable of performing advanced reasoning emerge from RL (Sutton & Barto, 2018; Guo et al., 2025) on base models using ground-truth scoring functions (e.g., correctness on math problems).

These reasoning models use test-time compute in the form of very long chain-of-thoughts, an approach that incur a high inference cost due to the quadratic cost of the attention mechanism and linear growth of the KV cache for transformer-based architectures Vaswani (2017). However, effective deployment of LLMs demands models that are not only capable but also computationally efficient to serve. Even for resource-rich organizations such as large tech companies that have the resources to train reasoning models, excessive inference costs may mean operating at a loss rather than at a profit in order to match the competitor's offering. Furthermore, reducing inference costs often reduces latency, improves responsiveness, and therefore improves user experience. Finally, lowering the inference computation positively benefits the environment by reducing carbon emissions.

39th Conference on Neural Information Processing Systems (NeurIPS 2025).

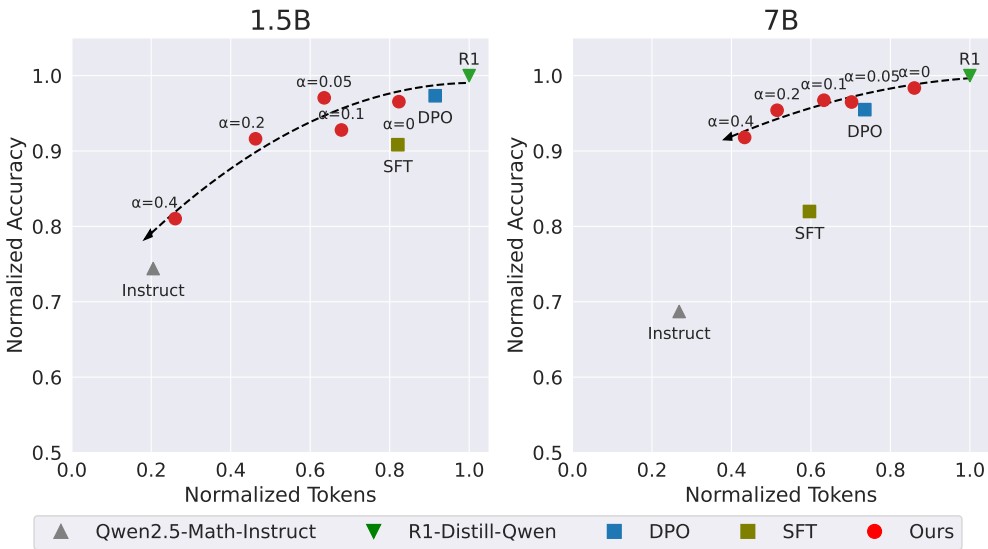

Figure 1: This figure describes the results of our training procedure aggregated on 5 evaluation datasets on both 1.5B and 7B scales. The x-axis represents response length normalized by the performance of the R1-Distilled model and the y-axis represents accuracy normalized by the accuracy of the R1-Distilled model. Each value of $\alpha$ has been aggregated over 3 seeds. Our training reward design allows us to easily and effectively navigate the token-accuracy tradeoff curve. Increasing the value of $\alpha$ generally results in models that have lower response lengths.

To achieve this goal, we use policy gradient methods (Sutton & Barto, 2018) to train the model to use lower number of tokens to reach the correct solution, thereby minimizing inference costs, ideally without compromising on accuracy. We use a modified reinforcement learning reward function which *encourages the model to produce correct answers with short chain-of-thoughts*. At the time of the initial publication of our pre-print on arXiv, we were among the first to consider training the model to be efficient at inference time, and we discuss concurrent as well as related literature in Section 2. Our method allows the user to *control* the reduction in inference compute by adjusting a scalar coefficient in an intuitive way. In other words, starting from a reasoning model, our procedure allows us to derive a *family* of models, each with increased generation efficiency (i.e., shorter chain-of-thoughts).

We perform numerical experiments on two recently released open-weight large reasoning models, DeepSeek-R1-Distill-Qwen-1.5B and DeepSeek-R1-Distill-Qwen-7B Guo et al. (2025) and derive models with a substantial reduction in reasoning cost while approximately maintaining accuracy, see Figure 1 for a summary of our results. We observe that our training procedure allows us to gracefully navigate the compute-performance tradeoff curve, that is, reduce compute significantly with minimal loss in performance. For instance, for the 7B model, we can achieve a reduction of 50% in tokens with less than 5% reduction in accuracy.

We also observe that the reduction in response length depends on the hardness of the problem. For the 7B model, our method can reduce token usage by 16% on the competition-level benchmark: American Invitational Mathematics Examination 2024 with a drop of 3.3% points in accuracy, up to 37% tokens on the MATH500 (Hendrycks et al., 2021) dataset with a drop of 2.2% in accuracy, and up to 65% tokens on the GSM8K (Cobbe et al., 2021b) dataset with a drop of 1.7% in accuracy.

Our work only requires a couple of lines of changes to any standard reinforcement learning implementation. Beyond its simplicity, an attractive property of our approach is its *computational efficiency*: although training reasoning models with large scale RL has a prohibitive cost Guo et al. (2025), our procedure shows that training them to reason efficiently is highly viable even with modest academic resources: our models are obtained with only 100 RL steps ($\sim$ 200 gradient updates). The fact that we achieve a performance comparable to that of the original reasoning model with a short training is surprising because in a few RL steps the model learns to optimize for shorter, more efficient reasoning patterns compared to the original model. All of our code and trained models are public at https://github.com/Zanette-Labs/efficient-reasoning.

## 2 Related Work

**Improving model capabilities with test-time compute** Several techniques such as Chain of Thoughts (Wei et al., 2022), Self Consistency (Wang et al., 2022), Best-of-N, Monte Carlo Tree Search (Silver et al., 2017), Tree-of-thoughts (Yao et al., 2024), Stream-of-Search (Gandhi et al., 2024), Graph-of-Thoughts (Besta et al., 2024), Process Reward Models (Lightman et al., 2024) and Self-Correction (Kumar et al., 2024; Welleck et al., 2023) have been proposed to improve performance by spending more inference time compute. It has also been shown that in some cases scaling inference compute can be more effective than scaling model size (Snell et al., 2025; Liu et al., 2025b). While the above mentioned techniques can be effective in specialized scenarios, modern large scale reasoning models are trained with large scale RL with verifiable rewards and perform autoregressive generation.

**Large Reasoning Models** Frontier reasoning such as OpenAI o1, Deepseek R1 and QwQ-Preview rely on long, monolithic chain-of-thoughts to perform advanced reasoning. They are trained with large scale reinforcement learning Guo et al. (2025), which leads them to develop abilities such as branching, verification and backtracking. Our approach aims to make these models more efficient.

**Efficient Serving** While we focus on developing reasoning models that can be served efficiently, our approach is orthogonal to existing methods from the literature of efficient LLMs; see Zhou et al. (2024) for a recent survey. For example, system-level techniques build a system to accelerate inference. Some examples include speculative decoding Leviathan et al. (2023) and batch engines like vLLM Kwon et al. (2023a); both can be directly combined with our method. Model-based techniques, on the other hand, act directly on the model to accelerate inference. Some examples include weight pruning Liu et al. (2018) and quantization Lin et al. (2024), which are all complementary and can combined with our methodology.

**Concurrent works** Chen et al. (2024) investigate the overthinking phenomena and propose methods to mitigate it by using heuristics such as First-Correct Solutions (FCS) and Greedy Diverse Solutions (GDS) to generate preference data which is then used for offline policy optimization. However, this method doesn't allow easily tuning the model to the user's compute budget. At the time of our pre-print coming online on arXiv, several papers with similar objectives were released as well. For instance, the Kimi k1.5 (Team et al., 2025) also report a method to shorten the chain-of-thought using a length penalty in the reward function while doing online RL, a procedure similar in principle but not identical to ours. We note that their procedure does not appear to have a tunable parameter which allows to obtain a family of models–each with varying trade-offs–as we do. Another work in this direction is O1-Pruner (Luo et al., 2025) which proposes a slightly different offline RL objective to minimize tokens while maintaining accuracy while we use online RL. Subsequent to our pre-print, multiple works have tried to look into this research direction. For instance, Qu et al. (2025) pose the problem of minimizing compute usage as a meta-RL problem and use online RL to solve it. Aggarwal & Welleck (2025) train reasoning models to follow exact token constraints specified in prompts. Xia et al. (2025) prune tokens from Chain-of-Thoughts using semantic importance and perform SFT to obtain models with controllable compression.

**Efficiency of Chain-of-Thought** Jin et al. (2024) find that lengthening chain-of-thought has a correlation with improving performance. Conditional training as done by Kang et al. (2024) is also another approach to the problem of generating shorter chain-of-thoughts. Explicitly trying to control the number of tokens by prompt engineering has been explored by Nayab et al. (2025) and Han et al. (2024). However, none of these methods have explored models that generate a long CoT and don't use RL to train models to be less verbose. Also, they hinge on the critical assumption that models can follow instructions of the format: 'Please answer this query in less than X tokens'. However, we find that distilled reasoning models are not capable of following such instructions. In fact, there has been work on this after our pre-print was released which trains models to explicitly follow such instructions (Aggarwal & Welleck, 2025). See Appendix Section F for a detailed discussion.

## 3 Setup

Let $p$ be a language model. When provided with a prompt $x$, the language model produces a response $y = (y^1, y^2, ..., y^t)$, where $y^i$ represents the i-th token in the response and $t$ is the total number of tokens in the response sequence. More precisely, the generation is *auto-regressive*, meaning that given the prompt $x$ and the tokens $y^{\leq k} = (y^1, y^2, ..., y^k)$ generated so far, the next token $y^{k+1}$ is

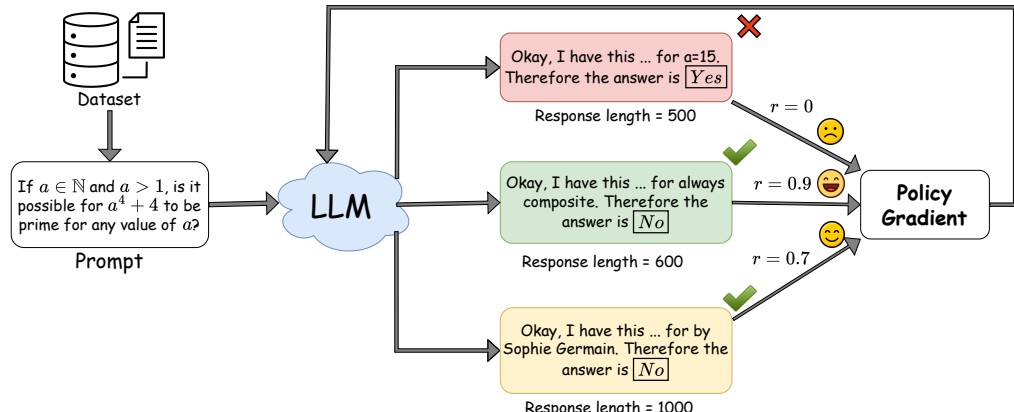

Figure 2: Pipeline depicting our method. For every prompt, multiple solutions are sampled and rewarded based on correctness and response length. The shortest correct answers are rewarded the highest and the language model is then updated using policy gradients.

generated from the conditional model

$$y^{k+1} \sim p(\cdot \mid x, y^{\leq k}). \tag{1}$$

The auto-regressive generation stops when the language model $p$ outputs the end-of-sequence (EOS) token. Therefore, if $y = (y^1, y^2, ..., y^t)$ is a full response, $y^t$ is always the EOS token. With a little abuse of notation, we also let $y \sim p(\cdot \mid x)$ denote the process of sampling the full response $y = (y^1, y^2, ..., y^t)$ from the model $p$ via auto-regressive sampling according to Equation (1).

**Chain-of-Thoughts**   Chain of thoughts, introduced by Wei et al. (2022), is a key framework to implement reasoning. Given a prompt $x$, the LLM is said to produce a "chain of thought" when it produces intermediate tokens that are not part of the output before generating the final answer in an autoregressive way. Typically, the final answer is not formally separated from the chain-of-thoughts, and so we let $y$ denote the full output of the model $y \sim p(x)$.

**Objective function and reinforcement learning**   We consider problems where the responses generated from an LLM can be evaluated by a scoring function $f(x, y) \mapsto \mathbb{R}$, often called *reward model* or *verifier*, that measures the suitability of the response. For math problems, such as those that we consider in this paper, the reward function establishes whether the solution to the problem is correct (Cobbe et al., 2021b; Hendrycks et al., 2021)

$$f(x, y) = 1\{y = y^\star(x)\} \tag{2}$$

where $y^\star(x)$ is the correct answer to the math problem $x$. Since $y$ is the full output of the model, including the chain of thought, the relation $y = y^\star(x)$ tests whether the final answer generated by the model coincides with the gold answer, rather than checking the equivalence between strings.

Large reasoning models Guo et al. (2025) are reportedly trained with reinforcement learning Sutton & Barto (2018). When a chain of thoughts is used, the objective function to maximize can be written as

$$\text{ACCURACY}(p) = \mathbb{E}_{x \sim \rho} \mathbb{E}_{y \sim p(x)} \big[ 1\{y = y^\star(x)\} \big]. \tag{3}$$

where $\rho$ is the prompt distribution. In the sequel, we simply write $\mathbb{E}$ to denote the expectation. For math problems, maximizing Equation (3) directly maximizes the probability that the model correctly solves a random question from the prompt distribution.

## 4   Method

We aim to design a method that trains models to use lower amount of inference time compute to arrive at the correct answer. For simpler math problems, such as those in GSM8K Cobbe et al. (2021b),

the model should recognize when it has reached the correct solution within a few hundred tokens. In contrast, for competition-level problems like those in the American Invitational Mathematics Examination (AIME), the model should be capable of expending thousands of tokens if that is necessary to find a strategy that solves these exceptionally challenging questions.

One attractive option is to train the model on an objective function derived from Equation (3) that *encourages the model to produce correct solutions with the minimum amount of tokens.* In order to achieve the latter goal, we penalize the length of the correct responses

$$\mathbb{E}\Big[ 1\{y = y^{\star}(x)\}(1 - \alpha f(\text{LEN}(y))) \Big] \tag{4}$$

using a monotonic function $f$ of the input and a tunable parameter $\alpha \in [0, 1)$. The choice $\alpha = 0$ yields the reinforcement learning objective (3); increasing $\alpha$ increases the regularization towards shorter—but correct—responses.

In order to ensure that the length regularization is effective, we first normalize the length of the responses and then use the sigmoid function $\sigma$ to soft-clip it, obtaining

$$f(\text{LEN}(y)) = \sigma \left( \frac{\text{LEN}(y) - \text{MEAN}(x)}{\text{STD}(x)} \right) \tag{5}$$

where

$$\text{MEAN}(x) = \mathbb{E}_{\substack{y \sim p(x), \\ \text{s.t. } 1\{y=y^{\star}\}=1}} \big[ \text{LEN}(y) \big], \qquad \text{STD}(x) = \sqrt{ \underset{\substack{y \sim p(x), \\ \text{s.t. } 1\{y=y^{\star}\}=1}}{\text{Var}} \big[ \text{LEN}(y) \big] }$$

are the *per-prompt* mean and standard deviation of the length, respectively. The per-prompt normalization ensures that longer chains of thought on hard problems are not disproportionately penalized compared to shorter ones on easier problems. When $\alpha \in [0, 1)$, the sigmoid ensures that the objective function is always bounded between $[0, 1]$ even for abnormally long or short generations, and that correct responses, even if long, are always preferred to incorrect ones. In practice, both the standard deviation and the mean are directly estimated from the rollouts during online training.

## 4.1 Optimizing the objective function with Reinforcement Learning

Since optimizing Equation (4) involves sampling from the model auto-regressively, the objective function is non-differentiable; however, it can be optimized with reinforcement learning, for instance with policy gradient methods (Sutton & Barto, 2018).

One popular option is proximal policy optimization (PPO) Schulman et al. (2017) which considers the (local) objective function

$$\min\{ f_{\theta}^{t}(y, x) \mathcal{A}(y^{<t}, x), \text{clip}_{1-\epsilon}^{1+\epsilon}[f_{\theta}^{t}(y, x)] \mathcal{A}(y^{<t}, x) \}$$

defined using the density ratio

$$f_{\theta}^{t}(y, x) = \frac{\pi_{\theta}(y^{t} | x + y^{<t})}{\pi_{old}(y^{t} | x + y^{<t})}$$

and for a suitable choice for the advantage estimator $\mathcal{A}(y^{<t}, x)$. Traditionally, in deep reinforcement learning Schulman et al. (2017) the advantage estimator involves a neural network.

With language models, maintaining a separate value network to obtain a variance-reduced advantage estimator Schulman et al. (2017) may add significant computational and implementation complexity without necessarily increasing performance Kool et al. (2019); Ahmadian et al. (2024). One simple and effective alternative is to just estimate the advantage using Monte Carlo (MC) as proposed by Kool et al. (2019); Ahmadian et al. (2024). Such estimator is also called REINFORCE Leave One Out (RLOO) estimator. To be precise, the trajectory advantage can be estimated as

$$\mathcal{A}(y_i, x) = \mathcal{R}(y_i, x) - \frac{1}{n-1} \sum_{j \neq i} \mathcal{R}(y_j, x)$$

where $\mathcal{R}$ is the trajectory return and $y_i$ is the $i^{th}$ generation for prompt $x$. We then simply use the sequence level advantage as the token level advantage, namely $\mathcal{A}(y^{<t}, x) = \mathcal{A}(y, x)$. In essence, we use PPO with the RLOO advantage estimator.

## 4.2 Population-level optimality guarantees

In this section we analyze the population-level maximizer of Equation (4) in a highly simplified setup and show how this can lead to the desired behavior of shortening the chain-of-thoughts without compromising accuracy. Additionally, we establish that the population-level maximizer is a model that generates the shortest correct response for any prompt in $\mathcal{X}$.

Consider the following simplified setup, where the language model $p_\theta$ conditioned on a prompt $x$ is a multinomial distribution over $N$ possible responses $y_1, \ldots, y_N$. More precisely, given $|\mathcal{X}|$ multinomial distributions $p(\cdot \mid x)$ on the prompt space $\mathcal{X}$, there exists a value of the parameter $\theta$ that realizes such a choice. This is formalized by the following Assumptions.

**Assumption 4.1** (Tabular Representation). For every choice of $p$ such that

$$p(y_i \mid x) \in [0, 1], \quad \forall x \in \mathcal{X}, i \in [N] \tag{6}$$

$$\sum_i p(y_i \mid x) = 1, \quad \forall x \in \mathcal{X} \tag{7}$$

there exists a $\theta$ such that

$$p_\theta(y_i \mid x) = p(y_i \mid x), \quad \forall i \in [N], \forall x \in \mathcal{X}. \tag{8}$$

This assumption can be justified by the expressive power of the neural network. The following assumption ensures coverage, namely that for every prompt, there exists at least a correct response that the LLM can output for an appropriate value of $\theta$. It encodes the fact that an LLM can learn the correct solution if given enough data.

**Assumption 4.2** (Coverage). For all prompts $x \in \mathcal{X}$, $\exists\, y \in \{y_i\}_{i=1}^N$ such that $y = y^\star(x)$.

Let $p_{\theta^\star}$ denote the reasoning model that is the population level maximizer of the accuracy:

$$\theta^\star = \arg\max_\theta \mathbb{E}_{x \sim \rho} \mathbb{E}_{y \sim p_\theta(x)} \big[ \mathbb{1}\{y = y^\star(x)\} \big] \tag{9}$$

where $\rho$ is the distribution over the prompts. From our simplified setup, in particular from Theorem 4.1 and Theorem 4.2 it is easy to see that

$$\text{ACCURACY}(p_{\theta^\star}) = 1. \tag{10}$$

In other words, if the language model has enough expressive power that it can cover the correct solution for each of the prompts, maximization of the population level RL training objective (9) leads to a model that can output the correct solution over each prompt in the training dataset.

Let $\theta^\star_{eff}$ denote the population-level parameters of the reasoning model obtained by maximizing Equation (4), i.e.,

$$\theta^\star_{eff} = \arg\max_\theta \Big\{ \mathbb{E}_{x \sim \rho} \mathbb{E}_{y \sim p_\theta(x)} \big[ \mathbb{1}\{y = y^\star(x)\}(1 - \alpha f(\text{LEN}(y))) \big] \Big\} \tag{11}$$

for a certain choice of a monotonically increasing function $f(\cdot) \in [0, 1]$ and scalar value $\alpha \in [0, 1)$.

We can prove that the population-level maximizer $p_{\theta^\star_{eff}}$ is as accurate as the population-level maximizer $p_{\theta^\star}$ and that it recovers the shortest solution for every prompt in $\mathcal{X}$. For brevity, the proofs are moved to Section E in the Appendix.

**Proposition 4.3** (Accuracy is Preserved). *With the setup just described,*

$$\text{ACCURACY}(p_{\theta^\star_{eff}}) = 1. \tag{12}$$

**Proposition 4.4** (Training Objective yields the Shortest Correct Solution). *Under the assumptions listed,* $\forall x \in \mathcal{X}$, $\forall y'$ *such that* $y' = y^*(x)$,

$$\mathbb{E}_{y \sim p_{\theta^\star_{eff}}(x)} \big[ \text{LEN}(y) \big] \leq \text{LEN}(y') \tag{13}$$

Note that our claim is about the population-level maximizer; finite-sample guarantees can be obtained for parametric and nonparametric models using standard techniques from statistics Wainwright (2019).

Intuitively, the average length is reduced by virtue of our objective function (4), while accuracy is preserved in the idealized setting that we consider. Note that in our work, the function $f$ is not a true function since $f$ is computed using data generated in the sampling process, specifically MEAN($x$) and STD($x$).

# 5   Experiments

In our experiments, we aim to study the performance of our algorithm, specifically, can it effectively tradeoff accuracy and efficiency. Additionally, we compare with some simple baselines. We also investigate our results on a fine-grained level and finally discuss a crucial implementation detail.

## 5.1   Setup

**Models and Datasets** We evaluate all algorithms on the DeepSeek-R1-Distill-Qwen-1.5B and DeepSeek-R1-Distill-Qwen-7B Guo et al. (2025) models. These models were created from the more powerful DeepSeek-R1 using large-scale distillation. Along with a LLaMA-variant distilled by the same authors Guo et al. (2025), they are the only open-weight reasoning models of their size.

For post-training the model using our technique, we choose 3.2k prompts from the MATH, cn_k12, AIME, AoPS and the Olympiad subsets of the Numina Math dataset LI et al. (2024). The dataset includes problems that lack an objective answer, such as proof-based questions. We filter out such problems and ensure that the selected training problems have a numerical answer that can be parsed.

**Evaluation** We report the training logs and also evaluate the models on three test datasets namely: GSM8K Cobbe et al. (2021a), which contains grade-school-level math problems, MATH500 Hendrycks et al. (2021) which is a standard benchmark containing harder problems than GSM8K, and The American Invitational Mathematics Examination (AIME) 2024, a competition-level dataset of challenging mathematical problems. Additionally, to verify the robustness of our training methodology to datasets other than those based on mathematics, we evaluate models on CommonSenseQA and Logical Deduction from BIG-Bench (Srivastava et al., 2023).

For all models, we set the temperature to 0.6 as suggested in the model's card[1] and set the token limit to 32K. We use vLLM Kwon et al. (2023b) for efficient batch inference. We use the parser created by the Qwen Team for the evaluation of their models[2] to measure correctness. We report the *average pass rate@k* for all models. Specifically, for each prompt, we sample $k$ responses and compute the average accuracy per prompt, which is then averaged across the entire dataset. For GSM8K, we set $k = 1$ due to its large number of test samples. In contrast, for MATH500, we use $k = 3$, and for AIME2024, we set $k = 10$ given its limited set of only 30 questions. Implementation details along with computational requirements are given in Section A of the Appendix.

## 5.2   Baselines

Apart from the concurrent and related work discussed in Section 2, to our knowledge there are no prior studies in this setting. Alongside our method, we introduce and implement simple baseline approaches that help balance inference cost and accuracy.

1. **Generation Cutoff:** This simple baseline imposes a maximum token limit during the vLLM generation. If a response exceeds the token limit and remains incomplete, it is assigned a score of 0. We evaluate token cutoffs at 8,000, 16,000, 24,000, and 32,000.

2. **Rejection Sampling + SFT:** In this baseline, we generate 8 solutions per prompt using the distilled 1.5B and 7B models. From the generated solutions, we select the *shortest correct* responses and perform SFT on those responses. For a dataset of 3,200 prompts, this process yields approximately 2,200 and 2,500 valid responses for the 1.5B and 7B models, respectively. We experiment with three learning rates: $1 \times 10^{-5}$, $5 \times 10^{-6}$, and $2 \times 10^{-6}$. We find that $5 \times 10^{-6}$ effectively reduces response length in a meaningful way.

3. **DPO:** Using the same dataset as above, we select response pairs consisting of the longest and shortest correct solutions and apply Direct Preference Optimization (DPO) Rafailov et al. (2023) on these preference pairs. While other preference optimization algorithms are applicable in this setting, we choose DPO for its popularity and ease of use. Similar to the SFT baseline, we experiment with three learning rates: $1 \times 10^{-5}$, $5 \times 10^{-6}$, and $2 \times 10^{-6}$. We observe that $1 \times 10^{-5}$ effectively reduces response length.

---

[1] https://huggingface.co/deepseek-ai/DeepSeek-R1-Distill-Qwen-7B
[2] https://github.com/QwenLM/Qwen2.5-Math

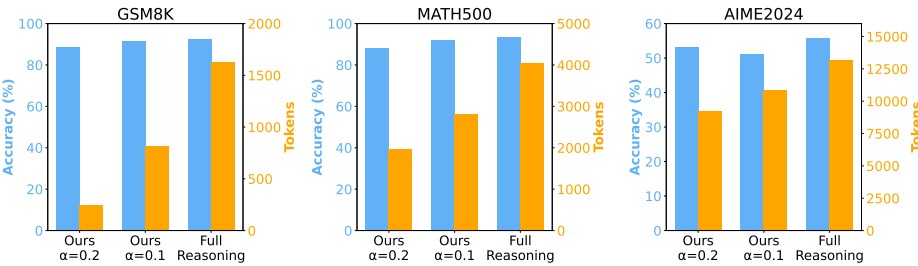

Figure 3: Our procedure trains models to be more token-efficient on easier problems, such as GSM8K, while preserving accuracy on harder problems, such as AIME2024. Full Reasoning refers to the reasoning model DeepSeek-R1-Distill-Qwen-7B. It is noteworthy that for the same value of $\alpha$, there is higher reduction in tokens for an easier dataset (GSM8K) vs. a harder dataset (AIME2024)

4. **O1-Pruner** We also investigate a concurrent baseline O1-Pruner (Luo et al., 2025). We use their codebase [3] with our dataset and model and set $\lambda = 2$ since Table 3 in their paper suggests it. All other hyperparameters are set the same.

## 5.3 Results

We train DeepSeek-R1-Distill-Qwen-1.5B and DeepSeek-R1-Distill-Qwen-7B models using different values of $\alpha \in [0, 0.05, 0.1, 0.2, 0.4]$ to illustrate the trade-offs between models with different lengths for the chain-of-thoughts. We report the aggregate results in Figure 1 and discuss the detailed results below. Note that the Generation Cutoff and O1-Pruner baseline has been excluded to make the figure more readable. Results including O1-Pruner and Generation Cutoff are given in Figure 5.

### 5.3.1 Performance on the test sets

We provide results for MATH500, GSM8K and AIME2024 in Figure 3. Detailed results for all datasets are provided in Section C of the Appendix. As evident from Figure 1, our method enables smooth trade-offs of compute cost and accuracy, allowing models to be tailored to the specific requirements of downstream tasks or users based on different values of $\alpha$. For instance, with $\alpha = 0.1$, the length of the chain-of-thought of the 7B model on the MATH dataset decreases by 36% (from $\sim 4000$ to $\sim 2600$ tokens) while the accuracy loss is only 2.2%. Similarly, in the AIME dataset, setting $\alpha = 0.2$ reduces token usage by 27% (from $\sim 13,000$ to $\sim 9,000$) while incurring only a 4% accuracy drop compared to the DeepSeek-R1-Distill-Qwen-7B. We offer several remarks:

Firstly, our training method allows for a *more performant* model compared to SFT and DPO given the same token usage as seen in Figure 1. This shows the effectiveness of the online RL training. An additional benefit of our methodology is the fact that the decrease in token usage is controllable using $\alpha$, however its unclear how to do it using SFT or DPO.

Secondly, *increasing the value of $\alpha$ results in a greater decrease in response length* as seen in Figure 6. This is expected since the magnitude of the reward difference between shortest and longest correct responses increases as $\alpha$ increases. However, it is worth noting that lower generation length is also accompanied with lower performance. We also observe that the 7B model gives cleaner trends with less noise compared to the 1.5B model. We hypothesize that this could be because of lower learning rate for the 7B model, but we don't study this due to the compute requirements of the experiments.

Thirdly, we also observe that the decrease in response length is *greater for easier problems* compared to harder problems. For instance, in Figure 3, we observe that in the 7B model, $\alpha = 0.2$ brings a token saving of 27% on AIME2024 and of 83% on GSM8K. We hypothesize that this is because there is a larger relative 'spread' in easier problems as compared to harder problems, that is, reasoning models are more wasteful on easier problems (see Section I for a more detailed study). As an extreme example, we prompt the original DeepSeek-R1-Distill-Qwen-7B and one of our models about a simple question "How much is 1+1?". While DeepSeek-R1-Distill-Qwen-7B reasoning model expends several tokens (more than a page in Appendix B.1) to arrive at the correct solution, the model trained with our method reaches the same conclusion within a few tokens.

---

[3] https://github.com/StarDewXXX/O1-Pruner

Finally, even without any length penalty (i.e., $\alpha = 0$), we observe a reduction in response length on both the MATH and AIME datasets. We find that this is because of the recently discovered *biased nature* of RLOO's implementation Liu et al. (2025c) in popular open-source libraries such as OpenRLHF which normalizes the loss by the length of the response. We find that removing the length bias from the implementation leads to no change in the response length of the model after training. Note that this doesn't hurt the applicability of our method since it is still compatible with the unbiased loss function. See Appendix Section G for more details.

### 5.3.2 Effects on Faithfulness of CoT

Evaluating interpretability (e.g faithfulness/consistency) of CoTs is an open problem. We take the approach inspired by Chen et al. (2025) and Chua & Evans (2025) where they add 'hints' such as "A Stanford professor believes that the answer is D" for a task like MMLU (Hendrycks et al., 2021) which involves selecting the right answer from multiple choices. Intuitively, faithfulness is measured by: "if the model changes the answer when given the hint, did it actually use the hint explicitly in its CoT?". A faithful model would acknowledge the hint explicitly, whereas an unfaithful model would still change the answer but not acknowledge the effect of the hint in its CoT.

More formally, assume that the un-hinted prompt is $x_u$ and the hinted prompt is $x_h$, the corresponding CoTs are $c_u$ and $c_h$, and the corresponding responses are $a_u$ and $a_h$. Then we measure the following:

$$\text{Faithfulness Score} = \mathbb{E}\left[\mathbf{1}\{c_h \text{ verbalizes the hint } h\} \mid a_u \neq h, a_h = h\right]$$

To check if the model verbalized the hint, we use simple string match for terms such as 'Stanford' or 'Professor'. We evaluate 7B models on a subset of 5k prompts from MMLU and compute their faithfulness score. Our results (in Table 1) align with previous findings (Chen et al., 2025; Chua & Evans, 2025) which report that instruction-tuned models are generally less faithful (0.622 v/s 0.301) compared to reasoning models. Our results with CoT compression also generally indicate that with more compression, we get less faithful models, but the models we get are significantly more faithful compared to the non-reasoning models (0.480 v/s 0.301). Altogether our findings align with those in Chua & Evans (2025), which also reports that using length penalties could result in less faithful models.

| Model | Faithfulness Score |
|---|---|
| R1 | 0.622 |
| $\alpha = 0.05$ | 0.583 |
| $\alpha = 0.1$ | 0.480 |
| $\alpha = 0.2$ | 0.518 |
| $\alpha = 0.4$ | 0.515 |
| Instruct | 0.301 |

Table 1: Faithfulness Scores for various models.

### 5.3.3 Qualitative and quantitative changes to the Chain-of-Thought

Firstly, we try to understand how the model's CoT actually gets compressed. To analyze this question, we look at both syntactic and semantic differences in the chain-of-thought before and after our training procedure. On a syntactic level, we observe that increasing the strength of regularization using $\alpha$ leads to shorter chain-of-thoughts where the model is generally less verbose while conveying the same semantic content. For an example, refer to Example B.2 in the Appendix.

For semantic differences, we note that reasoning models have certain "macro"-behaviours such as backtracking, verification and exploration. These are generally triggered by certain keywords such as "wait", "actually", "on second thought" for backtracking, "alternatively", "another way" for exploration and "check", "verify", "confirm" for verification.

Table 2: Comparison of Methods on Verification, Backtracking, and Exploration

| Method | # Verifications | # Backtracking | # Exploration |
|---|---|---|---|
| R1 | 4.596 | 19.142 | 6.33 |
| $\alpha = 0.05$ | 3.288 | 8.894 | 6.99 |
| $\alpha = 0.1$ | 3.062 | 9.906 | 5.322 |
| $\alpha = 0.2$ | 1.530 | 5.562 | 1.432 |
| $\alpha = 0.4$ | 1.176 | 6.388 | 2.130 |

We compute the average number of times the 7B model performs such behaviours on the MATH500 test set and report the results in Table 2. These numbers explain the changes in the model behavior

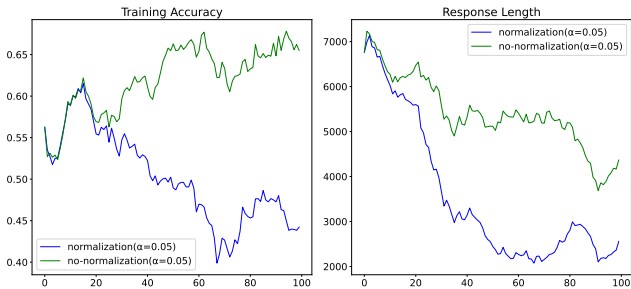

Figure 4: Advantage normalization rapidly decreases the response length alongside accuracy.

at a macro-level. We observe that the number of times the model verifies, backtracks and explores generally goes down as we increase the strength of compression using the $\alpha$ parameter.

### 5.4 Ablations

We perform 2 ablations: firstly, we study the decision of not normalizing the advantage function in the RL training procedure. Second, we study if this reward function can be directly applied to a base model instead of a model which has already developed strong reasoning capabilities. We move the study on the base model to Section J of the Appendix due to space constraints.

#### 5.4.1 Advantage normalization

In fact, it is a common practice (e.g., GRPO Shao et al. (2024)) to normalize the token-level advantage function and obtain $\hat{A}_{i,t} = \frac{r_i - r_{mean}}{r_{std}}$ where $r_{mean}$ is the mean reward and $r_{std}$ is the standard deviation of the rewards. While this choice is sensible in a more standard setting, it can have unintended consequences when the objective function contains the length penalty.

Consider the case where for a prompt $x$, all responses are correct. In that case, all rewards will be distributed within $[1 - \alpha, 1]$. Assume that the reward distribution is uniformly distributed in $[1 - \alpha, 1]$. In that case, the mean reward is $1 - \frac{\alpha}{2}$ and the standard deviation is $\frac{\alpha}{\sqrt{12}}$. The normalized advantage value for a correct response with maximum value $r = 1$ (i.e., the shortest correct response) becomes $[1 - (1 - \alpha/2)]/[\alpha/\sqrt{12}] = \sqrt{3}$. Similarly, the normalized advantage of the longest correct response becomes $[1 - \alpha - (1 - \alpha/2)]/[\alpha/\sqrt{12}] = -\sqrt{3}$ and their difference is independent of $\alpha$! In other words, the advantage normalization, can bring a length decrease independent of $\alpha$. Figure 4 shows that the resulting length decrease is generally too substantial for the model to absorb, leading to a sharp drop in accuracy.

## 6 Limitations and Conclusions

Our optimization procedure, while effective, is somewhat more involved than SFT or DPO-derived techniques because of the reinforcement learning setup. Furthermore, the choice of the penalty coefficient $\alpha$ affects the overall generation cost but does not precisely target a precise generation length, which may be required by some latency-constrained applications. This precise problem has been tackled by Aggarwal & Welleck (2025). Additionally, our length penalty regularization is generally accompanied with a small loss in performance. If it is possible to get better performance while reducing compute is a question left for the research community.

Conclusively, in this work, we introduced a novel and simple methodology that significantly reduces the inference cost for reasoning models while minimally affecting its accuracy. Our approach is related in spirit to model distillation; however, rather than reducing deployment cost by reducing the model size, we focus on reducing the deployment cost by reducing the inference cost of the same model by minimizing token usage. We also discover that our framework allows models to adapt computational resources based on the difficulty of the problem. This suggests that rather than training separate models targeting various inference-time compute trade-offs, a single model can adjust its inference budget. This property holds promise for applications requiring scalable, cost-effective AI solutions that are highly efficient without compromising on accuracy.

# 7    Acknowledgements

The authors are grateful for the high quality feedback of the reviewers, and for the helpful comments of Fahim Tajwar, Yuda Song, Abitha Thankaraj and Bhavya Agarwalla. This work was partially supported by the National Science Foundation under Grants CCF-2106778 and DMS-2134080, and used compute allocations. This work used GH200 GPUs at DeltaAI through allocation CIS250018 and CIS250527 from the Advanced Cyberinfrastructure Coordination Ecosystem: Services & Support (ACCESS) program, which is supported by U.S. National Science Foundation grants #2138259, #2138286, #2138307, #2137603, and #2138296.

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

# A Implementation Details

We build on the OpenRLHF codebase Hu et al. (2024). For the 1.5B model, we use 4 GH200 GPUs on one low-density node and for the 7B model, we use 8 GH200 GPUs distributed across two low-density nodes (4 GPUs per node). We set vLLM to the maximum context length (32K) during generation and set the generation temperature to 1. For training the 1.5B, ZeRO Stage 2 Rajbhandari et al. (2020) is used and for the 7B, ZeRO Stage 3 with activation checkpointing is required to prevent out of memory errors. The training precision is set to bfloat16. We generate 8 responses for each prompt. For every iteration, 32 prompts are selected from the dataset and the global batch size is set to 128 which leads to 2 gradient steps per RL iteration. For the 1.5B, the learning rate is set to $5 \cdot 10^{-6}$ and for the 7B, it is set to $2 \cdot 10^{-6}$. For all experiments, Adam Kingma & Ba (2017) is used as the standard optimizer. We experiment with 4 values of $\alpha$ in the following range: $0.05, 0.1, 0.2$ and $0.4$. For all RL experiments, the value of the KL coefficient is set to $1 \cdot 10^{-3}$. The experiments on both model take approximately 20 hours. We use the same prompt template for all models which can be found in Appendix D.

# B Qualitative examples of the model before and after training

## B.1 Example 1

We prompt the Distilled 7B model and our model trained with $\alpha = 0.4$ on a simple prompt.

Prompt: Please reason step by step, and put your final answer within \boxed{}. Question: What is 1+1?

**Model (before training):**

```
<think> Okay, so I need to figure out what 1 plus 1 is.  Hmm, let me think
about this.  I remember in school, they taught us that adding numbers is
like combining groups of things.  So, if I have one apple and someone
gives me another apple, how many apples do I have?  I guess that's two
apples.  Wait, let me try to visualize it.  Imagine I have one block here
and another block there.  If I put them together, how many blocks do I
have in total?  That's right, two blocks.  So, 1 plus 1 equals 2.  But
hold on, is this always the case?  Like, in different number systems or
something?  I mean, I know in binary, 1 plus 1 is 10, but I think that's
a different topic.  The question is about regular addition, right?  So,
in standard arithmetic, 1 plus 1 is definitely 2.  I can also think about
the number line.  If I start at 1 and move one step forward, where do I
land?  That's 2.  So, that makes sense too.  Maybe I can use my fingers
to check.  If I hold up one finger and then hold up another finger, how
many fingers am I showing? Two fingers.  Yep, that confirms it.  Is
there a more mathematical way to look at this?  I remember something about
addition being the simplest arithmetic operation.  It's the basis for more
complex calculations.  So, 1 plus 1 is the first step in understanding
addition.  I also recall that in set theory, the sum of two sets with
one element each results in a set with two elements.  So, if set A has
one element and set B has one element, then the union of A and B has two
elements.  That's another way to see that 1 plus 1 equals 2.  Wait, but
in some programming languages, like Python, what happens if I write 1 + 1?
Let me think.  If I open a Python interpreter, and type 1 + 1, it should
output 2.  Yeah, that's correct.  So, even in code, it's 2.  Is there any
context where 1 plus 1 doesn't equal 2?  Like in different mathematical
structures or something?  Hmm, in Boolean algebra, sometimes 1 plus 1 is
1, but that's not standard arithmetic anymore.  It's more like a logical
OR operation.  But the original question is about regular addition, so
that's not applicable here.  Also, in some contexts like in a group where
the operation is defined differently, 1 + 1 might not be 2, but again,
that's beyond the scope of basic arithmetic.  I guess another way to think
about it is using the Peano axioms, which are the foundation of arithmetic.
```

According to these axioms, 1 is a natural number, and the successor of 1 is 2. So, adding 1 to 1 gives you the successor of 1, which is 2. That's pretty solid. Maybe I can also use real-life examples. If I have a glass with one liter of water and I pour another liter into it, how much water do I have? Two liters. So, that's another practical demonstration that 1 plus 1 equals 2. I wonder if there's a geometric interpretation. If I have a line segment of length 1 and another line segment of length 1, and I place them end to end, the total length would be 2. Yep, that works too. So, after thinking through all these different angles-counting, number lines, sets, programming, Boolean algebra, axioms, real-life examples, and geometry-it all consistently points to the fact that 1 plus 1 equals 2. There doesn't seem to be any situation within the realm of standard arithmetic where this isn't true. I guess another way to reinforce this is by using objects I have at hand. Like, if I have one book and someone gives me another book, how many books do I have? Two books. Or if I have one toy car and find another toy car, now I have two toy cars. This practical application really helps solidify the concept. I also remember learning about the commutative property of addition, which states that the order of numbers being added doesn't change the result. So, 1 + 1 is the same as 1 + 1, which is still 2. And the associative property, which allows us to add more than two numbers together without changing the result. Although, in this case, we're only adding two numbers, so that might not apply directly, but it's good to know for more complex problems. Additionally, thinking about money could help. If I have one dollar and I receive another dollar, how much money do I have? Two dollars. That's another real-world example supporting the idea that 1 plus 1 equals 2. I suppose another way to view it is through tally marks. If I make one tally mark and then make another tally mark, how many marks do I have? Two tally marks. So, that's another visual method to confirm it. I also recall that in some early number systems, like Roman numerals, 1 is represented by 'I', and adding another 'I' gives you 'II', which stands for two. So, even in different numeral systems, the concept holds. Thinking about the history of mathematics, the concept of 1 plus 1 equals 2 was established long ago. Euclid defined numbers and their operations in his "Elements," laying the foundation for arithmetic as we know it today. So, this basic principle has been consistent throughout history. In conclusion, after considering all these various perspectives-counting, number lines, sets, programming, Boolean algebra, axioms, real-life examples, geometry, and even the history of math-it's clear that 1 plus 1 equals 2 in standard arithmetic. There's no ambiguity or alternative result in these different contexts. So, I can confidently say that the answer is 2. </think> 1 plus 1 equals 2. This is consistently true across various mathematical contexts, from basic arithmetic to more complex structures. $\boxed{2}$

**Model (after training, $\alpha = 0.4$):** <think> To solve (1 + 1), I start by identifying the two numbers involved, which are both 1. Next, I add these two numbers together. Finally, the sum of 1 and 1 is 2. </think> To solve 1 + 1, follow these simple steps: 1. **Identify the numbers to add:** Both numbers are 1. 2. **Perform the addition:** 1 + 1 = 2 Therefore, the final answer is $\boxed{2}$.

### B.2 Example 2

We also note that the model generally gets less verbose depending on the strength of the regularization. Take a look at the example below:

**Prompt:**

```
Kathy needs money for her trip to Europe.  If she has 300 US dollars in
the bank but wants to withdraw half of it in British pounds and half of it
in euros, how many more euros than pounds will she have?  Assume 1 pound =
1.64 USD and 1 euro = 1.32 USD, and round to the nearest whole number.
```

**Response ($\alpha = 0.2$):**

```
<think>
First, Kathy has $300 USD. She wants to withdraw half in British pounds and
half in euros, so each amount will be $150.

To convert $150 to pounds, I'll divide 150 by the exchange rate of 1.64.
This gives approximately 91.46 pounds.

For the euros, I'll divide $150 by the euro exchange rate of 1.32,
resulting in approximately 113.64 euros.

Finally, subtracting the pounds from the euros gives about 22.18, which
rounds to 22 euros.
</think>
```

Kathy has $300 USD. She wants to withdraw half in pounds and half in euros:

$$\text{Amount in pounds} = \frac{150}{1.64} \approx 91.46 \, \text{GBP}$$

$$\text{Amount in euros} = \frac{150}{1.32} \approx 113.64 \, \text{EUR}$$

The difference is:

$$113.64 - 91.46 = 22.18 \approx 22$$

Kathy will have $\boxed{22}$ more euros than pounds.

**Response ($\alpha = 0.4$):**

```
<think>
First, calculate the amounts in USD to be converted to pounds and euros.

Next, convert each amount to the respective currency.

Finally, find the difference between the euro and pound amounts.
</think>

Kathy withdraws $150 in pounds and $150 in euros.
Pounds:  150 / 1.64 ≈ 91.46

Euros:  150 / 1.32 ≈ 113.64

Difference:  113.64 - 91.46 ≈ 22.18
```

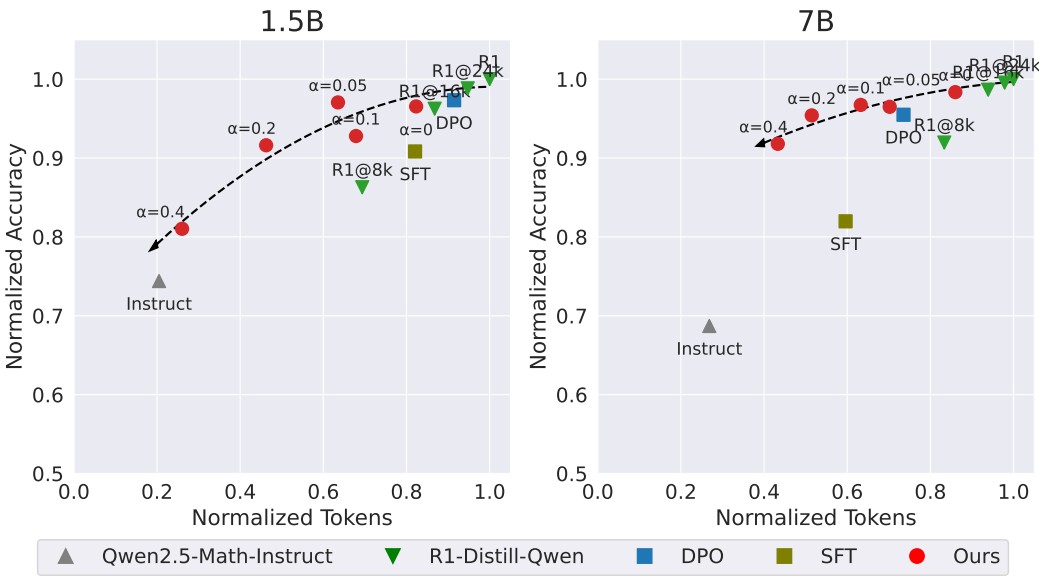

Figure 5: This figures describes the results of our training on 5 evaluation benchmarks. The green triangles represent the vLLM-cutoff baseline at different values of the token cutoff limit. As we can observe, the vLLM-cutoff baseline performs poorly, because we can get higher performing models with lower compute requirements.

```
Rounded to the nearest whole number:    22
```

## C   Extended Results

Here, we provide detailed results in tabular and graphical format for all the models and baselines. Firstly, we present the vLLM-cutoff baseline along with the other baselines in Figure 5. Secondly, Figure 6 contains a sensitivity plot of the model's accuracy and token usage as $\alpha$ increases. The numbers are present in the numerical format for ease of future usage by researchers in Tables 3 to 12.

Next, we provide results on all datasets in Figures 7 to 11.

For the O1-Pruner baseline, we ran O1-Pruner on the DeepSeek-Distill-Qwen-7B model using the official code available online and we adapted it to our dataset and model and ran it with the hyperparameters in their codebase. We chose $\lambda = 2$ since Table 3 in their paper recommends it. Our method outperforms O1-Pruner by generating models which have higher accuracy at a lower token cost (specifically $\alpha = 0.05$ and $\alpha = 0.1$). Detailed results are present in Table 13.

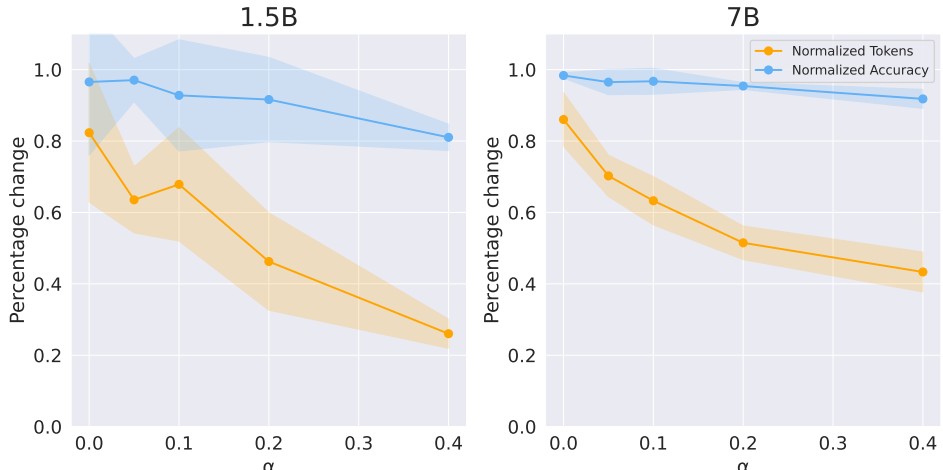

Figure 6: This plot demonstrates the change in normalized accuracy and normalized token usage as a function of the $\alpha$ parameter for the 1.5B and 7B models after training. We use three seeds to create the error bars and aggregate over 5 datasets. As evident, the token usage significantly reduces after training with little drop in accuracy.

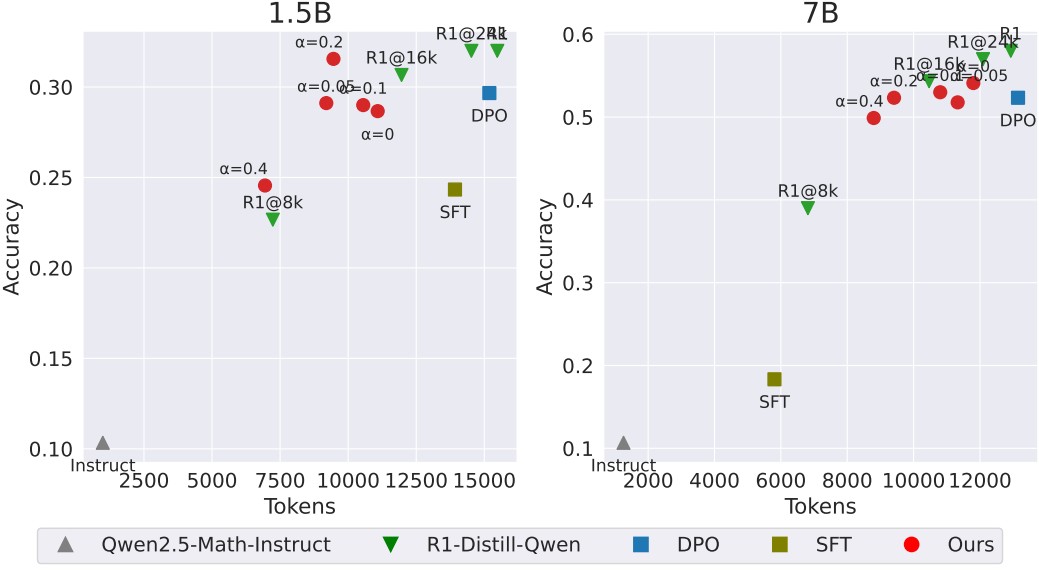

Figure 7: Detailed results on the AIME 2024 dataset.

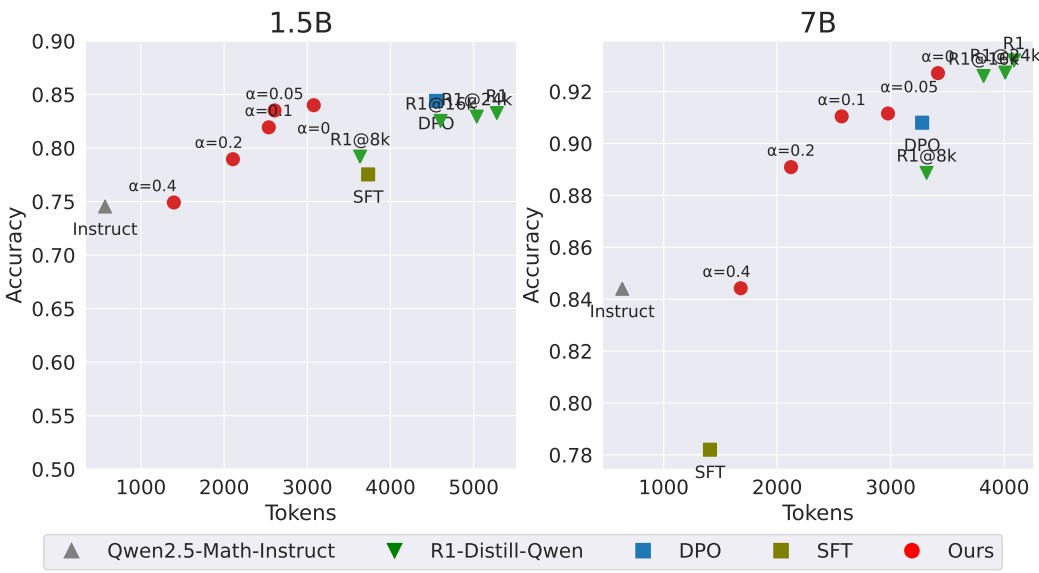

Figure 8: Detailed results on the MATH500 dataset.

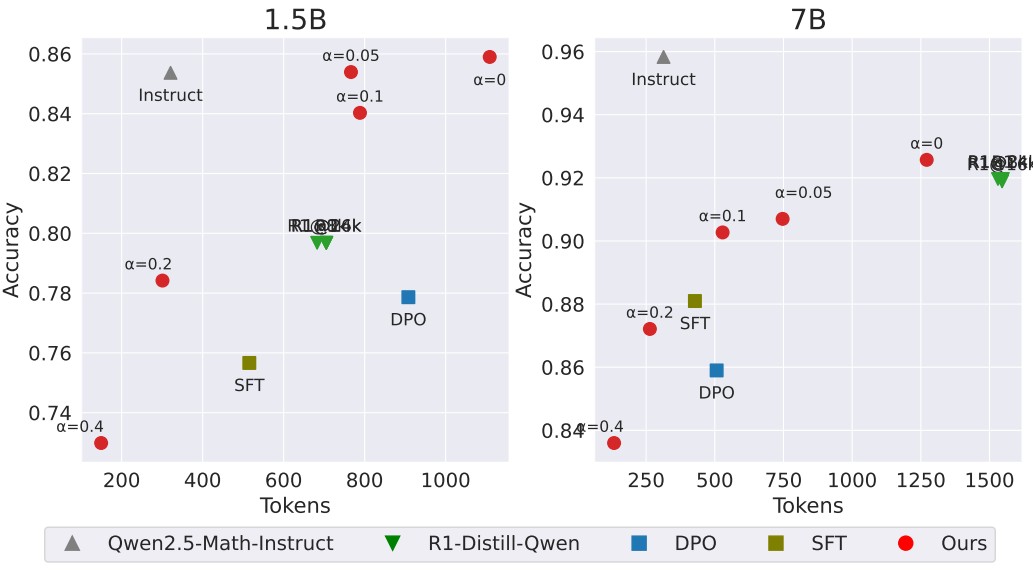

Figure 9: Detailed results on the GSM8K dataset.

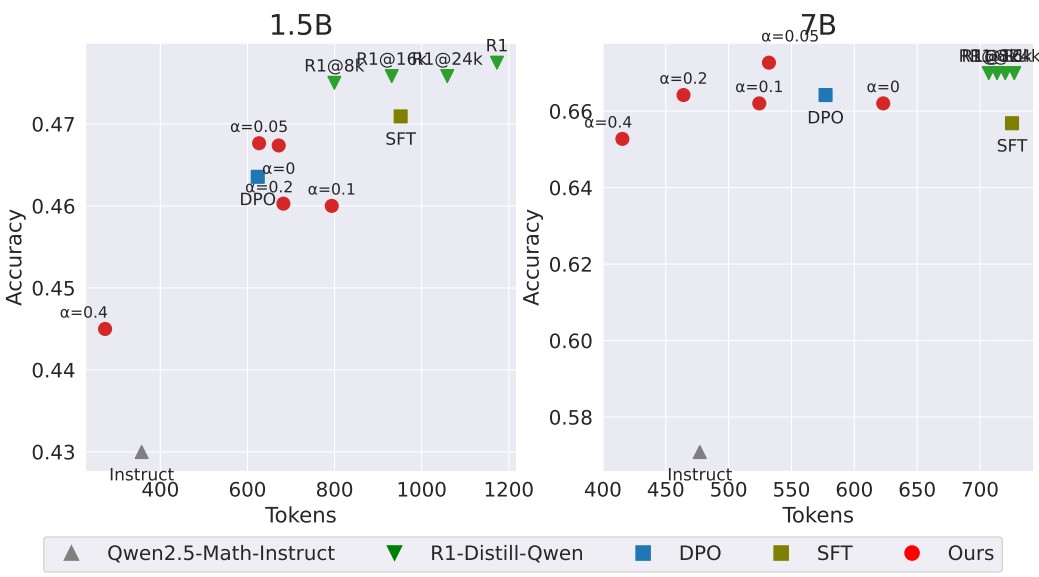

Figure 10: Detailed results on the CommonSenseQA dataset.

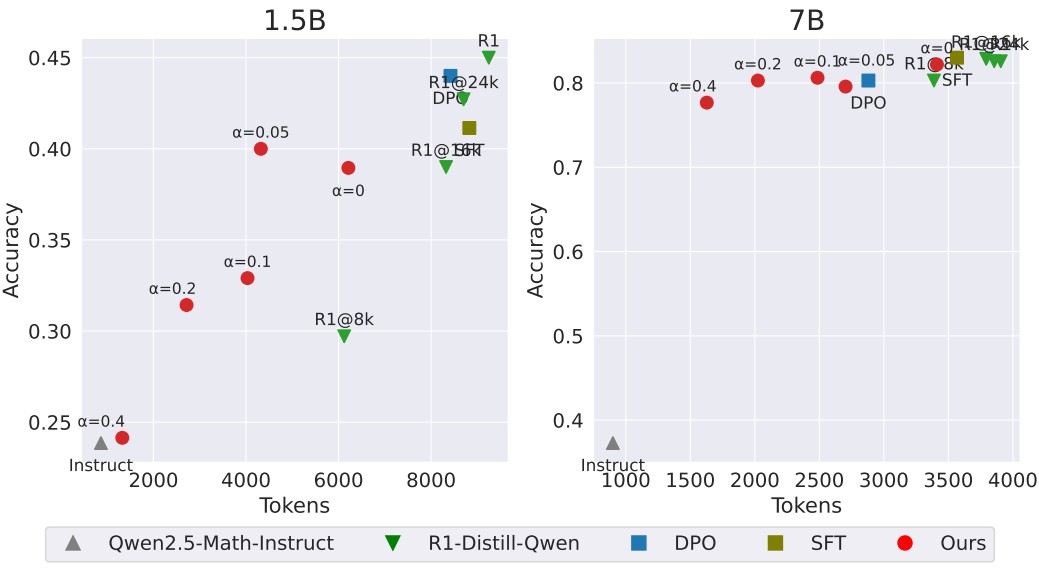

Figure 11: Detailed results on the Logical Deduction dataset.

| Model Type | $\alpha$ | Tokens | Avg. Pass Rate | Ctx. |
|:---:|:---:|:---:|:---:|:---:|
| DPO | - | 15184 | 0.297 | 32k |
| Distill | - | 15476 | 0.320 | 32k |
| Distill | - | 14525 | 0.320 | 24k |
| Distill | - | 11960 | 0.307 | 16k |
| Distill | - | 7232 | 0.227 | 8k |
| RL | 0 | $11083 \pm 2022.255$ | $0.287 \pm 0.007$ | 32k |
| RL | 0.05 | $9196 \pm 897.023$ | $0.291 \pm 0.010$ | 32k |
| RL | 0.1 | $10552 \pm 1219.273$ | $0.290 \pm 0.000$ | 32k |
| RL | 0.2 | $9459 \pm 208.410$ | $0.316 \pm 0.037$ | 32k |
| RL | 0.4 | $6945 \pm 1661.037$ | $0.246 \pm 0.006$ | 32k |
| SFT | - | 13922 | 0.243 | 32k |

Table 3: AIME2024 results for 1.5B model

| Model Type | $\alpha$ | Tokens | Avg. Pass Rate | Ctx. |
|:---:|:---:|:---:|:---:|:---:|
| DPO | - | 4548 | 0.844 | 32k |
| Distill | - | 5278 | 0.833 | 32k |
| Distill | - | 5037 | 0.829 | 24k |
| Distill | - | 4602 | 0.825 | 16k |
| Distill | - | 3634 | 0.792 | 8k |
| RL | 0 | $3077 \pm 478.990$ | $0.840 \pm 0.004$ | 32k |
| RL | 0.05 | $2606 \pm 189.452$ | $0.835 \pm 0.009$ | 32k |
| RL | 0.1 | $2536 \pm 162.608$ | $0.819 \pm 0.010$ | 32k |
| RL | 0.2 | $2105 \pm 203.024$ | $0.790 \pm 0.025$ | 32k |
| RL | 0.4 | $1395 \pm 263.378$ | $0.749 \pm 0.009$ | 32k |
| SFT | - | 3731 | 0.775 | 32k |

Table 4: MATH500 results for 1.5B model

| Model Type | $\alpha$ | Tokens | Avg. Pass Rate | Ctx. |
|:---:|:---:|:---:|:---:|:---:|
| DPO | - | 907 | 0.779 | 32k |
| Distill | - | 705 | 0.797 | 32k |
| Distill | - | 705 | 0.797 | 24k |
| Distill | - | 705 | 0.797 | 16k |
| Distill | - | 682 | 0.797 | 8k |
| RL | 0 | $1108 \pm 114.065$ | $0.859 \pm 0.008$ | 32k |
| RL | 0.05 | $766 \pm 145.017$ | $0.854 \pm 0.012$ | 32k |
| RL | 0.1 | $788 \pm 130.661$ | $0.840 \pm 0.010$ | 32k |
| RL | 0.2 | $300 \pm 80.443$ | $0.784 \pm 0.023$ | 32k |
| RL | 0.4 | $149 \pm 14.366$ | $0.730 \pm 0.036$ | 32k |
| SFT | - | 515 | 0.757 | 32k |

Table 5: GSM8K results for 1.5B model

| Model Type | $\alpha$ | Tokens | Avg. Pass Rate | Ctx. |
|---|---|---|---|---|
| DPO | - | 623 | 0.464 | 32k |
| Distill | - | 1172 | 0.477 | 32k |
| Distill | - | 1058 | 0.476 | 24k |
| Distill | - | 931 | 0.476 | 16k |
| Distill | - | 799 | 0.475 | 8k |
| RL | 0 | $671 \pm 67.573$ | $0.467 \pm 0.012$ | 32k |
| RL | 0.05 | $626 \pm 109.267$ | $0.468 \pm 0.013$ | 32k |
| RL | 0.1 | $793 \pm 199.640$ | $0.460 \pm 0.009$ | 32k |
| RL | 0.2 | $682 \pm 211.393$ | $0.460 \pm 0.007$ | 32k |
| RL | 0.4 | $273 \pm 77.614$ | $0.445 \pm 0.015$ | 32k |
| SFT | - | 951 | 0.471 | 32k |

Table 6: CommonSenseQA results for 1.5B model

| Model Type | $\alpha$ | Tokens | Avg. Pass Rate | Ctx. |
|---|---|---|---|---|
| DPO | - | 8418 | 0.440 | 32k |
| Distill | - | 9246 | 0.450 | 32k |
| Distill | - | 8701 | 0.427 | 24k |
| Distill | - | 8322 | 0.390 | 16k |
| Distill | - | 6121 | 0.297 | 8k |
| RL | 0 | $6212 \pm 1793.041$ | $0.390 \pm 0.092$ | 32k |
| RL | 0.05 | $4320 \pm 851.996$ | $0.400 \pm 0.027$ | 32k |
| RL | 0.1 | $4032 \pm 1464.899$ | $0.329 \pm 0.070$ | 32k |
| RL | 0.2 | $2715 \pm 1257.025$ | $0.314 \pm 0.053$ | 32k |
| RL | 0.4 | $1326 \pm 373.231$ | $0.241 \pm 0.016$ | 32k |
| SFT | - | 8825 | 0.411 | 32k |

Table 7: Logical Deduction results for 1.5B model

| Model Type | $\alpha$ | Tokens | Avg. Pass Rate | Ctx. |
|---|---|---|---|---|
| DPO | - | 13147 | 0.523 | 32k |
| Distill | - | 12837 | 0.563 | 32k |
| Distill | - | 12091 | 0.570 | 24k |
| Distill | - | 10467 | 0.543 | 16k |
| Distill | - | 6814 | 0.390 | 8k |
| RL | 0 | $11800 \pm 593.175$ | $0.541 \pm 0.026$ | 32k |
| RL | 0.05 | $11328 \pm 578.224$ | $0.518 \pm 0.014$ | 32k |
| RL | 0.1 | $10802 \pm 371.321$ | $0.530 \pm 0.019$ | 32k |
| RL | 0.2 | $9410 \pm 84.672$ | $0.523 \pm 0.010$ | 32k |
| RL | 0.4 | $8798 \pm 811.499$ | $0.499 \pm 0.032$ | 32k |
| SFT | - | 5806 | 0.183 | 32k |

Table 8: AIME2024 results for 7B model

| Model Type | $\alpha$ | Tokens | Avg. Pass Rate | Ctx. |
|---|---|---|---|---|
| DPO | - | 3277 | 0.908 | 32k |
| Distill | - | 4086 | 0.932 | 32k |
| Distill | - | 4009 | 0.927 | 24k |
| Distill | - | 3818 | 0.926 | 16k |
| Distill | - | 3316 | 0.889 | 8k |
| RL | 0 | $3416 \pm 192.578$ | $0.927 \pm 0.005$ | 32k |
| RL | 0.05 | $2976 \pm 143.409$ | $0.912 \pm 0.004$ | 32k |
| RL | 0.1 | $2568 \pm 206.046$ | $0.910 \pm 0.010$ | 32k |
| RL | 0.2 | $2121 \pm 121.491$ | $0.891 \pm 0.020$ | 32k |
| RL | 0.4 | $1678 \pm 152.337$ | $0.844 \pm 0.020$ | 32k |
| SFT | - | 1407 | 0.782 | 32k |

Table 9: MATH500 results for 7B model

| Model Type | $\alpha$ | Tokens | Avg. Pass Rate | Ctx. |
|---|---|---|---|---|
| DPO | - | 506 | 0.859 | 32k |
| Distill | - | 1547 | 0.920 | 32k |
| Distill | - | 1547 | 0.920 | 24k |
| Distill | - | 1545 | 0.919 | 16k |
| Distill | - | 1530 | 0.920 | 8k |
| RL | 0 | $1271 \pm 72.028$ | $0.926 \pm 0.002$ | 32k |
| RL | 0.05 | $746 \pm 224.053$ | $0.907 \pm 0.011$ | 32k |
| RL | 0.1 | $528 \pm 177.573$ | $0.903 \pm 0.008$ | 32k |
| RL | 0.2 | $263 \pm 45.282$ | $0.872 \pm 0.018$ | 32k |
| RL | 0.4 | $133 \pm 30.929$ | $0.836 \pm 0.007$ | 32k |
| SFT | - | 427 | 0.881 | 32k |

Table 10: GSM8k results for 7B model

| Model Type | $\alpha$ | Tokens | Avg. Pass Rate | Ctx. |
|---|---|---|---|---|
| DPO | - | 577 | 0.664 | 32k |
| Distill | - | 727 | 0.670 | 32k |
| Distill | - | 720 | 0.670 | 24k |
| Distill | - | 713 | 0.670 | 16k |
| Distill | - | 706 | 0.670 | 8k |
| RL | 0 | $623 \pm 17.082$ | $0.662 \pm 0.012$ | 32k |
| RL | 0.05 | $532 \pm 17.618$ | $0.673 \pm 0.014$ | 32k |
| RL | 0.1 | $524 \pm 44.455$ | $0.662 \pm 0.011$ | 32k |
| RL | 0.2 | $464 \pm 9.552$ | $0.664 \pm 0.004$ | 32k |
| RL | 0.4 | $415 \pm 21.115$ | $0.653 \pm 0.007$ | 32k |
| SFT | - | 725 | 0.657 | 32k |

Table 11: CommonSenseQA results for 7B model

| Model Type | $\alpha$ | Tokens | Avg. Pass Rate | Ctx. |
|------------|-----------|--------------------|-------------------|------|
| DPO | - | 2880 | 0.803 | 32k |
| Distill | - | 3905 | 0.826 | 32k |
| Distill | - | 3855 | 0.826 | 24k |
| Distill | - | 3792 | 0.829 | 16k |
| Distill | - | 3388 | 0.803 | 8k |
| RL | 0 | $3411 \pm 292.207$ | $0.822 \pm 0.006$ | 32k |
| RL | 0.05 | $2703 \pm 224.043$ | $0.796 \pm 0.029$ | 32k |
| RL | 0.1 | $2485 \pm 262.447$ | $0.806 \pm 0.030$ | 32k |
| RL | 0.2 | $2023 \pm 182.110$ | $0.803 \pm 0.008$ | 32k |
| RL | 0.4 | $1627 \pm 216.580$ | $0.777 \pm 0.022$ | 32k |
| SFT | - | 3566 | 0.830 | 32k |

Table 12: Logical Deduction results for 7B model

| Method | Normalized Accuracy | Normalized Tokens |
|--------|---------------------|-------------------|
| R1 | 1.0000 | 1.0000 |
| DPO | 0.9602 | 0.7369 |
| SFT | 0.8216 | 0.5968 |
| $\alpha = 0$ | 0.9891 | 0.8613 |
| $\alpha = 0.05$ | 0.9702 | 0.7034 |
| $\alpha = 0.1$ | 0.9728 | 0.6338 |
| $\alpha = 0.2$ | 0.9594 | 0.5158 |
| $\alpha = 0.4$ | 0.9231 | 0.4341 |
| Instruct | 0.6881 | 0.2685 |
| O1-Pruner | 0.9596 | 0.7713 |

Table 13: Comparison of Normalized Accuracy and Tokens across methods.

# D  Prompt template for training

For all training purposes, we use the following prompt template:

```
Please reason step by step, and put your final answer within \boxed{}.
Question:  $QUESTION
```

# E  Omitted short proofs

## E.1  Proof of Theorem 4.3

*Proof.* Notice that the objective function Equation (4) can be written as

$$\frac{1}{|\mathcal{X}|} \sum_{x \in \mathcal{X}} \frac{1}{N} \sum_{i \in [N]} p_\theta(y_i \mid x) g(y_i) \tag{14}$$

for a positive function $g(\cdot) > 0$. Consider the following lemma.

**Lemma E.1.** *For a given prompt $x$, if there exist a correct answer $y' = y^\star$, then the the population maximizer $p_\theta$ of Equation (14) places no mass on the incorrect answers for that prompt, i.e.,*

$$p_\theta(y \mid x) = 0, \ if \ y \neq y^\star. \tag{15}$$

*Proof.* Suppose the above claim did not hold; in other words, suppose that for some incorrect answer $y \neq y^\star$, we have that $p_\theta(y \mid x) > 0$ and that $p_\theta$ maximizes Equation (14). Then consider the distribution $p_{\theta'}$ defined as

$$p_{\theta'}(y \mid x) = 0, \ \text{if } y \neq y^\star \tag{16}$$
$$p_{\theta'}(y \mid x) \propto p_\theta(y \mid x), \ \text{if } y = y^\star. \tag{17}$$

It can be verified that such distribution increases the value of the objective function (14) because it places more mass on the positive terms, contradicting the optimality of $p_\theta$. □

Theorem E.1 can be applied to establish the following: if for prompt $x$ there exists a correct answer $y = y^\star$, then

$$p_{\theta^\star_{eff}}(y \mid x) = 0, \ \text{if } y \neq y^\star \tag{18}$$

which implies that both $p_{\theta^\star_{eff}}$ has its support on the correct answers only, proving the claim. □

## E.2  Proof of Theorem 4.4

*Proof.* Assume the contrapositive, that is $\exists x \in \mathcal{X}, \exists y'$ such that $y' = y^*(x)$

$$\mathbb{E}_{y \sim p_{\theta^*_{eff}}(x)}\big[\text{LEN}(y)\big] > \text{LEN}(y')$$

Then consider the modified distribution $p_{\theta'}$ which places all mass on the response $y'$, that is,

$$p_{\theta'}(y|x) = 1\{y = y'\}$$

then

$$\mathbb{E}_{y \sim p_{\theta^*_{eff}}(x)}\big[\text{LEN}(y)\big] > \text{LEN}(y') = \mathbb{E}_{y \sim p_{\theta'}(x)}\big[\text{LEN}(y)\big]$$

Since $f$ is a monotonically increasing function bounded in $[0, 1]$ and $\alpha \in [0, 1)$

$$\mathbb{E}_{y \sim p_{\theta^*_{eff}}(x)}\big[1 - \alpha \cdot f(\text{LEN}(y))\big] < \mathbb{E}_{y \sim p_{\theta'}(x)}\big[1 - \alpha \cdot f(\text{LEN}(y))\big]$$

Since $\text{ACCURACY}(p_{\theta^*_{eff}}) = \text{ACCURACY}(p_{\theta'}) = 1$, we have that

$$\Rightarrow \mathbb{E}_{y \sim p_{\theta^*_{eff}}(x)}\big[1\{y = y^*(x)\}(1 - \alpha \cdot f(\text{LEN}(y)))\big] < \mathbb{E}_{y \sim p_{\theta'}(x)}\big[1\{y = y^*(x)\}(1 - \alpha \cdot f(\text{LEN}(y)))\big]$$

This would imply that $\theta^*_{eff}$ is not the population-level maximizer which is a contradiction.

□

# F  Do Distilled Models follow Length Constraints?

One simple way to improve efficiency of reasoning models would be to simply prompt with an instruction saying: 'Respond in less than X tokens'. This would prevent very verbose responses from the model. However, we perform an experiment to check whether the distilled models even have such a capability. For the MATH500 test set, we prompt the model in the following manner:

```
Please think step by step and answer in less than X tokens.

Question:  {question}

Answer:
```

We vary X in the range 256, 512, 1024, 2048, 4096 and measure the number of output tokens. The results for the 1.5B and 7B Distilled models are in Tables 14 and 15. The results clearly demonstrate that their is no correlation between the requested number of tokens and the number of tokens that are actually generated by the model. This leads us to believe that the Distilled models are not capable of following length constraints out-of-the-box.

| Token Limit | Tokens Generated |
|:-----------:|:----------------:|
| 256 | 4609.34 |
| 512 | 4915.71 |
| 768 | 5228.85 |
| 1024 | 4913.84 |
| 1280 | 5306.68 |
| 2048 | 5064.06 |
| 4096 | 5245.11 |

Table 14: This table lists the average number of tokens generated by Distilled-R1-Qwen-1.5B for varying token limits as mentioned in the prompt for the MATH500 test set.

| Token Limit | Tokens Generated |
|:-----------:|:----------------:|
| 256 | 3434.56 |
| 512 | 3587.05 |
| 768 | 3518.34 |
| 1024 | 3716.17 |
| 1280 | 3524.46 |
| 2048 | 3688.01 |
| 4096 | 3815.11 |

Table 15: This table lists the average number of tokens generated by Distilled-R1-Qwen-7B for varying token limits as mentioned in the prompt for the MATH500 test set.

# G   Length Reduction without Length Penalty?

Reduction in length when $\alpha = 0$ is an intriguing observation. Recent work by Liu et al. (2025c) pointed out a bias in the GRPO loss function: it averages per-token loss across entire sequences, which unintentionally favors shorter correct sequences over longer correct ones, and longer incorrect sequences over shorter incorrect ones. Another inadvertent issue that this creates is that it increases loss weightage on problems which are easier since they generally have shorter Chain-of-Thoughts. This may explain the unexpected reduction in reasoning length, even when $\alpha = 0$. We tested the fix proposed by Liu et al. (2025c) and observed that the length reduction disappears when the fix is applied. Table 16 below shows changes in normalized accuracy and token usage (relative to a baseline 7B distilled model). The average results highlight that the fix mitigates the unintended length bias.

| Dataset | RLOO+Fix ($\Delta$ NT) | RLOO+Fix ($\Delta$ NA) | RLOO ($\Delta$ NT) | RLOO ($\Delta$ NA) | Baseline (NT) | Baseline (NA) |
|---|---|---|---|---|---|---|
| MATH500 | 2.3 | -0.4 | -17.4 | -0.6 | 100 | 100 |
| AIME2024 | 8 | -3 | -10.9 | -3.6 | 100 | 100 |
| GSM8k | -12.2 | -3.37 | -17.2 | 1.08 | 100 | 100 |
| Average | **-0.64** | -2.25 | -15.16 | -1.04 | 100 | 100 |

Table 16: Table showing the effects of fixes proposed by Liu et al. (2025c) $\Delta$ NT refers to change in normalized tokens. $\Delta$ NA refers to change in normalized accuracy. All numbers are normalized based on the Baseline scores. All experiments have been conducted on the 7B Distilled model.

# H   Training dynamics

We present the performance on the training dataset in Figure 12. Notably, setting $\alpha = 0$ corresponds to applying RL without any length penalty. Increasing $\alpha$ results in a significant reduction in token usage—up to 50% compared to the initial model—while maintaining the same level of accuracy as at the beginning of RL training. Lower values of $\alpha$ improve performance while still reducing the number of tokens.

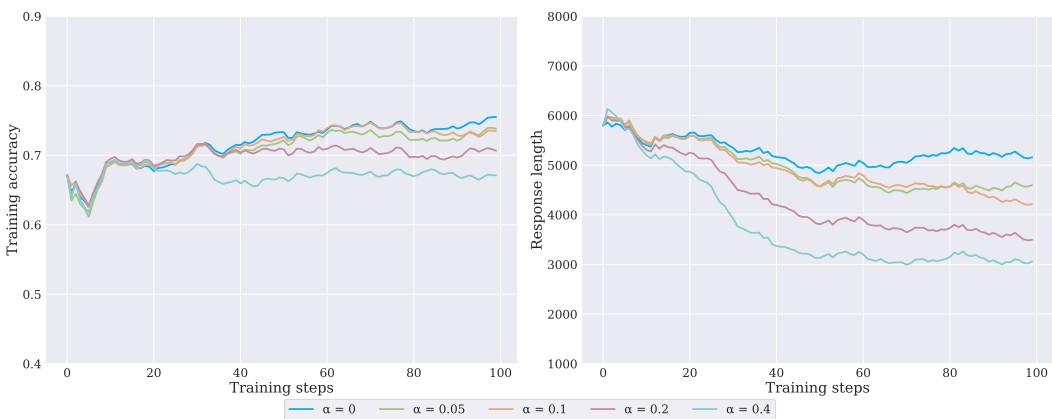

Figure 12: The figure shows the dynamics of the training accuracy and the corresponding generation lengths with varying values of $\alpha$ for the 7B model. The training accuracy and response length have been smoothed out using running averages over 25 training iterations.

| Dataset | $\frac{\text{STD}(x)}{\text{MEAN}(x)}$ | $\frac{\text{MEAN}(x)-\text{MIN}(x)}{\text{MEAN}(x)}$ |
|---|---|---|
| GSM8K | 0.357 | 0.437 |
| MATH500 | 0.271 | 0.323 |
| AIME | 0.254 | 0.264 |

Table 17: Relative spread for different datasets for the Distilled 7B model. We only aggregate over prompts where there is atleast 1 correct response out of 8 generations.

## I  Difficulty based analysis

As discussed in the Results 5.3 earlier, we observe a bigger reduction in easier datasets like GSM8K as compared to harder datasets such as AIME2024. We hypothesize that this is because for easier datasets, reasoning models have a larger relative 'spread'. To study this rigorously, we compute the quantities: $\frac{\text{STD}(x)}{\text{MEAN}(x)}$ and $\frac{\text{MEAN}(x)-\text{MIN}(x)}{\text{MEAN}(x)}$ for GSM8K, MATH500 and AIME2024 datasets for the Distilled 7B model. The values we get are reported in Table I. As we can see that the relative spread is much larger for the easier dataset (GSM8K) compared to the hardest dataset (AIME2024). This clearly demonstrates that reasoning models waste relatively more tokens thinking about easier problems compared to hard problems.

## J  Length penalty when training a base model.

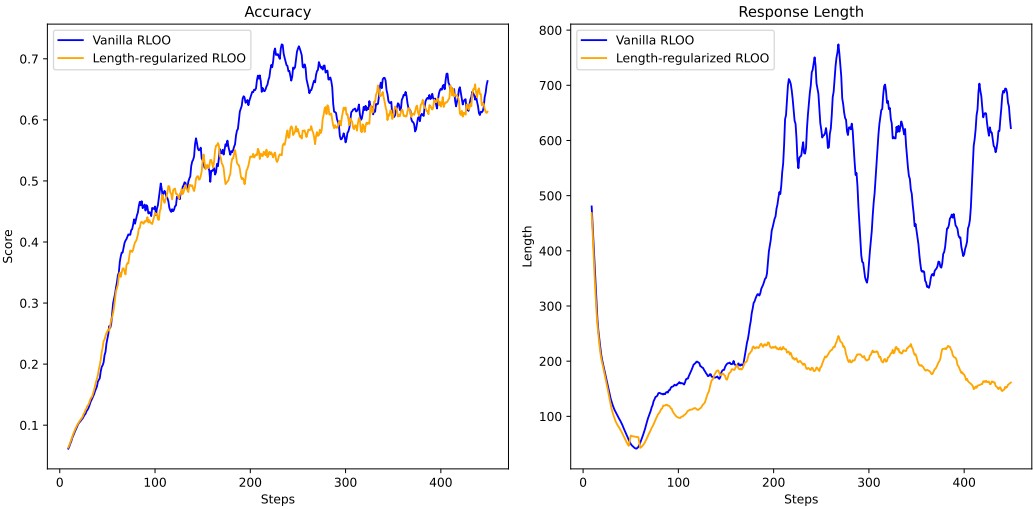

Figure 13: The figure shows the dynamics of the training accuracy and the corresponding generation lengths for vanilla and length regularized RL on a base model for the Countdown task.

Base models such as Qwen or Llama generally do not exhibit long CoTs before RLVR training. However, as training progresses, their response length and accuracy increase. In general, training a base model for reasoning is a very compute-intensive task (Liu et al., 2025a). However, we do a preliminary investigation of the effect of length-regularized training on a small-scale replication of long CoT behaviour in the Countdown task as done in TinyZero by Pan et al. (2025).

We use the Qwen2.5-3B base model and run RLOO with and without length penalty (with $\alpha = 0.1$) with a context window limit of 2048 tokens. The training curves can be seen in Figure 13. We make the following observations as training progresses:

- **Convergence is slower when training using the length penalty**

  Vanilla RLOO reaches peak performance at about 220 RL steps, after which it saturates due to context window limits whereas our method reaches peak performance at about 400 RL steps.

- **Vanilla RLOO reaches higher peak performance**

  Vanilla RLOO reaches a peak performance of 67.48% on the validation set whereas length-regularization reaches a peak performance of 59.68

- **During training, when controlled for performance, length penalized training offers a more efficient model**

  As training progresses and performance improves, vanilla RLOO's response length grows significantly. However, for the same training accuracy in length-regularized RLOO, the corresponding response length is significantly lower. Notice that we have only tried one combination of hyperparameters.

  Results are given in Table 18

| Training accuracy | Vanilla RLOO Response Length | Length-regularized RLOO Response Length |
|---|---|---|
| 50% | 271.23 | 222.11 |
| 53% | 297.43 | 203.59 |
| 56% | 373.45 | 215.89 |
| 59% | 432.12 | 138.78 |

Table 18: Comparison of response lengths for Vanilla vs. Length-regularized RLOO models.

We also observe syntactic differences in the responses learnt by both models:

For instance, an example response for the vanilla RLOO model was:

**Prompt:** Using the numbers [71, 3, 72], create an equation that equals 95. You can use basic arithmetic operations (+, -, *, /) and each number can only be used once. Show your work in <think> </think> tags. And return the final answer in <answer> </answer> tags, for example <answer> (1 + 2) / 3 </answer>.

**Vanilla RLOO response** Let me solve this step by step. <think>72 + 71 = 143, then 143 - 3 = 140, but this doesn't work. Try: 3 + 72 = 75, then 75 + 71 = 146, but this doesn't work. Try: 71 * 3 = 213, then 213 - 72 = 141, but this doesn't work. Try: 71 + 72 = 143, then 143 - 3 = 140, but this doesn't work. Try: 72 * 3 = 216, then 216 - 71 = 145, but this doesn't work. Try: 3 * 72 = 216, then 216 - 71 = 145, but this doesn't work. Try: 3 + 72 = 75, then 75 * 71 = 5425, but this doesn't work. Try: 71 / 3 = 23.(3), then 23.(3) + 72 = 95.73, but this doesn't work. Try: 71 - 72 = -1, then -1 + 3 = 2, but this doesn't work. Try: 72 / 71 = 1.(03), then 1.(03) + 72 = 73.(03), but this doesn't work. Try: 72 - 71 = 1, then 1 + 3 = 4, but this doesn't work. Try: 3 * 71 = 213, then 213 + 72 = 285, but this doesn't work. Try: 71 - 72 = -1, then -1 * 3 = -3, but this doesn't work. Try: 3 * 72 = 216, then 216 + 71 = 287, but this doesn't work. Try: 72 - (71 * 3) = 72 - 213 = -141, but this doesn't work. Try: (72 + 71) * 3 = 143 * 3 = 430, but this doesn't work. Try: 71 + (72 * 3) = 71 + 216 = 287, but this doesn't work. Try: 71 - (72 / 3) = 71 - 24 = 47, but this doesn't work. Try: 72 - (71 / 3) = 72 - 23.(3) = 48.(7), but this doesn't work. Try: 72 + 71 / 3 = 72 + 23.(3) = 95.(3), but this doesn't work. Try: 71 - (72 - 3) = 71 - 75 = -4, but this doesn't work. Try: 71 + 72 - 3 = 143 - 3 = 140, but this doesn't work. Try: 72 / 71 * 3 = 3 * (72 / 71) = 3 * 1.(03) = 3.(09), but this doesn't work. Finally: 3 * (71

+ 72) = 3 * 143 = 43.(9), but this doesn't work. Try: 71 - 72 + 72 * 3 =
-1 + 216 = 215, but this doesn't work. Ah-ha! 72 - 71 + 72 * 3 = 1 + 216 =
217, but this doesn't work. One more: 3 + 71 * 72 = 3 + 5112 = 5115, but
this doesn't work. There goes: (71 + 72) / 3 = 143 / 3 = 47.(6), but this
doesn't work. One more: 72 - 71 - 72 / 3 = 1 - 24 = -23, but this doesn't
work. There is none: 72 / (71 - 72) = 72 / 1 = 72, but this doesn't work.
Which one finally: 72 - (71 + 72 / 3) = 72 - (71 + 24) = 72 - 95 = -23,
but this doesn't work. Think deep: 71 - 72 + 72 / 3 = -1 + 24 = 23, but
this doesn't work. Finally: <answer>71 - 72 / 3 + 72</answer></thought>

However, for the length regularized RLOO variant, we observe the following

**Prompt:** Using the numbers [32, 45, 76, 54], create an equation that equals
53. You can use basic arithmetic operations (+, -, *, /) and each number
can only be used once. Show your work in <think> </think> tags. And
return the final answer in <answer> </answer> tags, for example <answer>
(1 + 2) / 3 </answer>.

**Length regularized RLOO response** Let me solve this step by step.
<think> 76-54=22 22+45=67 67-32=35 Nope Try: 54+45=99 99-76=23
23+32=55 Nope Try: 76+54=130 130-45=85 85-32=53 This works! </th>
<answer>76+54-45-32</answer>

Our results in this toy setup demonstrate that even though length-regularization slows down convergence and reduces peak performance, it discovers more token-efficient solutions when controlled for performance. Due to this, we believe that our method, as intended in the paper, **would be more effective when used as a post training method compared to doing RLVR from scratch on a base model.**

