# OpenReview forum: "Training Language Models to Reason Efficiently"
_NeurIPS.cc/2025/Conference — NeurIPS 2025 poster_

### Official Review · Reviewer_vDnZ · 2025-06-14

**Clarity:** 3
**Significance:** 3
**Originality:** 2
**Rating:** 4
**Confidence:** 3

**Summary:**

The main idea in this paper is to use RL (PPO) with an objective function to teach models to produce more succinct chain of thought while maintaining accuracy of the final response.  The overall idea is an interesting one, seems novel, and useful for deployment of models.  The work seems to demonstrate a drop in the tokens used (even for the hardest math datasets such as AIME), with some loss to accuracy.

**Questions:**

Please see above questions - especially about breaking down the results per dataset.

**Ethical Concerns:**

["NO or VERY MINOR ethics concerns only"]

**Final Justification:**

I think the paper reports an interesting result but is not yet ready for publication - the reasons are outlined in my comments, more empirical support is needed.

**Limitations:**

Yes

**Quality:**

3

**Strengths And Weaknesses:**

Strengths:
1.  Well written paper, straightforward application of standard RL techniques to reducing the length of the response while trying to maintain accuracy.
2.  Important as the cost of deploying so-called 'reasoning models' grows.
3.  Somewhat novel in terms of application of RL techniques to the problem of lengthy responses.

Weaknesses:
1.  The experimental section (and really its writeup) could be improved.  Specifically - the paper describes use of multiple math datasets (AIME, MATH500, GSM8K) and Logical Deduction and CommonSenseQA in its writeup - but presents results only the math datasets in Figure 3.  Figure 1 I think shows the results across all 5 datasets as an average but it would be better to collapse the space in Figures 1 and 3 to show the breakdown per dataset as in Figure 1.  Figure 3 does not add much more than what is in Figure 1.
2.  To what degree is the efficiency gains reported in terms of RL training a function of the specific model used?  Deepseek R1 has already been trained extensively on 'reasoning' steps using RL.  I wonder to what extent the efficiency reports here are because of the specific model here.  What if you started with a base model and tried to teach it to reason using the modified objective function?  Would you expect this efficiency?

---

> ### Author Rebuttal · Authors · 2025-07-31
>
> We thank the reviewer for their insightful questions and appreciate the fact that they found our paper simple to read and our method novel.
>
> Below, we address their concerns:
>
> 1. **Breakdown of results per dataset:**
>
> The results for every dataset individually can be found in Section C of the Appendix. We put them there because of a lack of space in the main paper. Figure 3 shows that the reduction in relative response length is dependent on the hardness of the dataset as we mention in line 292. We acknowledge that the presentation of Figure 3 could be improved to highlight the actual message. We will improve upon it in the final version of the paper.
>
> 2. **What if we start with a base model?**
>
> We thank the reviewer for asking the question.
>
> Base models such as Qwen or Llama generally do not exhibit long CoTs before RLVR training. However, as training progresses their response length and accuracy increase. In general, training a base model for reasoning is a very compute-intensive task[1]. However, we investigate the effect of length-controlled training on a small scale replication of long CoT behaviour in the Countdown task as done in TinyZero [2].
>
> We use the Qwen2.5-3B base model and run RLOO with and without length penalty (with $\alpha=0.1$) with a context window limit of 2048 tokens. We make the following observations as training progresses:
>
> a. **Convergence is slower when training using the length penalty**
>
> Vanilla RLOO reaches peak performance at about 220 RL steps, after which it saturates due to context window limits whereas our method reaches peak performance at about 400 RL steps.
>
> b. **Vanilla RLOO reaches higher peak performance**
>
> Vanilla RLOO reaches a peak performance of 67.48% on the validation set whereas length-regularization reaches a peak performance of 59.68%
>
> c. **During training, when controlled for performance, length penalized training offers a more efficient model**
>
> As training progresses and performance improves, vanilla RLOO’s response length grows significantly. However, for the same training accuracy in length-regularized RLOO, the corresponding response length is significantly lower. Notice that we have only tried one combination of hyperparameters.
>
> Results are given in the table below:
>
> | Training accuracy             | Vanilla RLOO Response Length       | Length regularized RLOO Response Length    |
> |---------------------|-----------------|------------------|
> | 50%                  | 271.23         | 222.11          |
> | 53%                 | 297.43          | 203.59          |
> | 56%                | 373.45         | 215.89           |
> | 59%               | 432.12          | 138.78          |
>
> We also observe syntactic differences in the responses learnt by both models:
>
> For instance, an example response for the vanilla RLOO model was:
>
> ```
> User: Using the numbers [71, 3, 72], create an equation that equals 95. You can use basic arithmetic operations (+, -, *, /) and each number can only be used once. Show your work in <think> </think> tags. And return the final answer in <answer> </answer> tags, for example <answer> (1 + 2) / 3 </answer>.
> ```
>
> ```
> Assistant: Let me solve this step by step. <think>72 + 71 = 143, then 143 - 3 = 140, but this doesn't work. Try: 3 + 72 = 75, then 75 + 71 = 146, but this doesn't work. Try: 71 * 3 = 213, then 213 - 72 = 141, but this doesn't work. Try: 71 + 72 = 143, then 143 - 3 = 140, but this doesn't work. Try: 72 * 3 = 216, then 216 - 71 = 145, but this doesn't work. Try: 3 * 72 = 216, then 216 - 71 = 145, but this doesn't work. Try: 3 + 72 = 75, then 75 * 71 = 5425, but this doesn't work. Try: 71 / 3 = 23.(3), then 23.(3) + 72 = 95.73, but this doesn't work. Try: 71 - 72 = -1, then -1 + 3 = 2, but this doesn't work. Try: 72 / 71 = 1.(03), then 1.(03) + 72 = 73.(03), but this doesn't work. Try: 72 - 71 = 1, then 1 + 3 = 4, but this doesn't work. Try: 3 * 71 = 213, then 213 + 72 = 285, but this doesn't work. Try: 71 - 72 = -1, then -1 * 3 = -3, but this doesn't work. Try: 3 * 72 = 216, then 216 + 71 = 287, but this doesn't work. Try: 72 - (71 * 3) = 72 - 213 = -141, but this doesn't work. Try: (72 + 71) * 3 = 143 * 3 = 430, but this doesn't work. Try: 71 + (72 * 3) = 71 + 216 = 287, but this doesn't work. Try: 71 - (72 / 3) = 71 - 24 = 47, but this doesn't work. Try: 72 - (71 / 3) = 72 - 23.(3) = 48.(7), but this doesn't work. Try: 72 + 71 / 3 = 72 + 23.(3) = 95.(3), but this doesn't work. Try: 71 - (72 - 3) = 71 - 75 = -4, but this doesn't work. Try: 71 + 72 - 3 = 143 - 3 = 140, but this doesn't work. Try: 72 / 71 * 3 = 3 * (72 / 71) = 3 * 1.(03) = 3.(09), but this doesn't work. Finally: 3 * (71 + 72) = 3 * 143 = 43.(9), but this doesn't work. Try: 71 - 72 + 72 * 3 = -1 + 216 = 215, but this doesn't work. Ah-ha! 72 - 71 + 72 * 3 = 1 + 216 = 217, but this doesn't work. One more: 3 + 71 * 72 = 3 + 5112 = 5115, but this doesn't work. There goes: (71 + 72) / 3 = 143 / 3 = 47.(6), but this doesn't work. One more: 72 - 71 - 72 / 3 = 1 - 24 = -23, but this doesn't work. There is none: 72 / (71 - 72) = 72 / 1 = 72, but this doesn't work. Which one finally: 72 - (71 + 72 / 3) = 72 - (71 + 24) = 72 - 95 = -23, but this doesn't work. Think deep: 71 - 72 + 72 / 3 = -1 + 24 = 23, but this doesn't work. Finally: <answer>71 - 72 / 3 + 72</answer></thought>
> ```
>
> But for the regularized model an example response looked like:
>
> ```
> User: Using the numbers [32, 45, 76, 54], create an equation that equals 53. You can use basic arithmetic operations (+, -, *, /) and each number can only be used once. Show your work in <think> </think> tags. And return the final answer in <answer> </answer> tags, for example <answer> (1 + 2) / 3 </answer>.
> ```
> ```
> Assistant: Let me solve this step by step. <think> 76-54=22 22+45=67 67-32=35 Nope Try: 54+45=99 99-76=23 23+32=55 Nope Try: 76+54=130 130-45=85 85-32=53 This works! </th>
> <answer>76+54-45-32</answer>
> ```
>
> Our results in this toy setup demonstrate that even though length-regularization slows down convergence and reduces peak performance, it discovers more token-efficient solutions when controlled for performance. Due to this, we believe that **our method, as intended in the paper, would be more effective when used as a post training method** compared to doing RLVR from scratch on a base model.
>
> We thank the reviewer again for asking the question, and we are happy to include a section about this experiment in the camera ready if the paper is accepted, which we believe will increase the quality of the manuscript.
>
> **References:**
>
> [1] ProRL: Prolonged Reinforcement Learning Expands Reasoning Boundaries in Large Language Models, arXiv 2025
>
> [2] TinyZero: Jiayi Pan and Junjie Zhang and Xingyao Wang and Lifan Yuan and Hao Peng and Alane Suhr

---

> > ### Comment · Reviewer_vDnZ · 2025-08-03
> >
> > Thank you for the explanation - I believe the paper would indeed be strengthened by the experiments you report.

---

### Official Review · Reviewer_3yKA · 2025-06-26

**Clarity:** 3
**Significance:** 3
**Originality:** 3
**Rating:** 4
**Confidence:** 4

**Summary:**

This paper proposes a simple training method to reduce the thinking length for reasoning models. The method considers generation length as part of the reward signals, and promotes generations that achieved correct answers with shorter lengths. The paper provides theoretical justification that such a training objective preserves accuracy and achieves the lowest generation lengths.

**Questions:**

1. While in general reducing reasoning length for efficiency improvement is desirable, enforcing shorter generations during training may also introduce some side effects, for example reward hacking. It would be great that the paper provides some analysis on the "correctness" of the reduced reasoning trajectories, making sure that the model was not taking advantage of artifacts to achieve correct answers. In fact, for some easy problems, generating some irrelevant phrases or not generating any thoughts at all can already achieve correct answers.

2. In many cases over-generation and under-generation can both hurt model performance, and there usually exist an "optimal" length of thoughts for specific problems (e.g. [1]). The proposal of this paper aggressively pursues the goal towards "under-generation", and I'm wondering whether relaxing the training objectives a bit may further improving model quality (for example choosing a different functional form for f(LEN(y)), to penalize under-generation too).

[1] When More is Less: Understanding Chain-of-Thought Length in LLMs, arXiv. 2025

3. Please consider comparing against more competitive baselines addressing the same problem of efficient inference scaling. See e.g. [2].

[2] Stop Overthinking: A Survey on Efficient Reasoning for Large Language Models, arXiv, 2025

**Ethical Concerns:**

["NO or VERY MINOR ethics concerns only"]

**Final Justification:**

The authors' responses properly addressed my questions. I will keep my original rating.

**Limitations:**

Yes.

**Quality:**

3

**Strengths And Weaknesses:**

Strengths:
- The paper addresses a very important problem of improving reasoning efficiency for LLMs, one of the most discussed topics recently.
- The proposed training objective modification is simple and easy to follow.
- Theoretical justification is provided to justify the effectiveness the proposed approach.

Weakness:
- Since efficient reasoning is a popular research topic and many approaches have been proposed, the baselines are kind of weak and the paper could have compared with more methods addressing the same problem.
- The proposed approach works for tasks where binary ground truth labels are available. It remains unclear whether it works effectively for cases where answer correctness cannot be easily decided (e.g. open-ended QA, deep research etc.)

---

> ### Author Rebuttal · Authors · 2025-07-31
>
> We thank the reviewer for their comments on the paper’s importance, simplicity and theoretical backing. We address their main concerns below:
>
> 1. **Baselines**
>
> We thank the reviewer's suggestion about running more baseline. As the reviewer noticed, there are several concurrent works, many of which also do not release code. Following the reviewers suggestions, we ran O1-Pruner [1] on the DeepSeek-Distill-Qwen-7B model using the official code available online and we adapted it to our dataset and model and ran it with the hyperparameters in their codebase. We chose $\lambda=2$ since Table 3 in their paper recommends it. Our method outperforms O1-Pruner by generating models which have higher accuracy at a lower token cost (specifically $\alpha=0.05$ and $\alpha=1$).  Detailed results are present in the table below:
>
> | Method              | Normalized Accuracy        | Normalized Tokens       |
> |---------------------|-----------------|------------------|
> | R1                  | 1.0000          | 1.0000           |
> | DPO                 | 0.9602          | 0.7369           |
> | SFT                 | 0.8216          | 0.5968           |
> | α = 0               | 0.9891          | 0.8613           |
> | α = 0.05            | 0.9702          | 0.7034           |
> | α = 0.1             | 0.9728          | 0.6338           |
> | α = 0.2             | 0.9594          | 0.5158           |
> | α = 0.4             | 0.9231          | 0.4341           |
> | Instruct            | 0.6881          | 0.2685           |
> | O1-Pruner     | 0.9596          | 0.7713           |
>
>
>
>
> 2. **Only works when binary ground truth is available**
>
> We acknowledge that this is a limitation and extending this to problems without binary rewards is definitely a question of open research. We will add this to our Limitations Section. One simple way to solve this problem would be to simply modify the reward function $r(x, y)$ for any task where the LLM generates responses to $r’(x, y) = r(x, y)*(1-f(LEN(y))$. This is a generalization of our method as well (where rewards are 0/1). Whether this is the right method for problems without binary rewards, is outside the scope of our current study.
>
>
> 3. **Reward hacking**
>
> *“for some easy problems, generating some irrelevant phrases or not generating any thoughts at all can already achieve correct answers.”*
>
> We thank the reviewer for raising this important concern regarding reward hacking. In our setup, the reward function is grounded in correctness (via ground-truth verification) and penalizes excessive length through a controllable factor. As a result, the only way to achieve a high reward is by producing responses that are both correct and concise. Generating irrelevant or off-topic content would consume tokens without improving the correctness signal, and is thus actively discouraged by the reward structure.
>
> Importantly, if a problem is simple enough that the model can provide a correct answer without any chain-of-thought reasoning, this is considered optimal under our objective—both in terms of efficiency and correctness. Nonetheless, we appreciate the reviewer’s suggestion and will clarify this point in the paper.
>
>
> 4. **Guiding towards optimal reasoning length**
>
> This is an excellent suggestion and there has been recent work which is related to this idea [2]. The key insight they have is that length penalties should only be applied when the pass rate for a prompt during online RL is high. This allows the model to compress its CoT only when it is highly confident.
>
>
> **References:**
>
> [1] O1-Pruner: Length-Harmonizing Fine-Tuning for O1-Like Reasoning Pruning, arXiv 2025
>
> [2] Just Enough Thinking: Efficient Reasoning with Adaptive Length Penalties Reinforcement Learning, arXiv 2025

---

> > ### Comment · Reviewer_3yKA · 2025-08-09
> >
> > Thank you authors for the detailed reply, they properly addressed my questions. I will keep my original rating.

---

### Official Review · Reviewer_YQoo · 2025-06-30

**Clarity:** 3
**Significance:** 3
**Originality:** 2
**Rating:** 4
**Confidence:** 3

**Summary:**

This paper proposes a method to train large language models to reason efficiently by reducing the length of their chain-of-thoughts during inference, thereby lowering computational costs. The core idea involves using policy gradient methods with a modified reinforcement learning reward function that penalizes response length while encouraging correct answers. A tunable hyperparameter, `α`, allows users to control the trade-off between token usage and accuracy. The authors demonstrate their approach on 1.5B and 7B DeepSeek-R1-Distill-Qwen models across math and reasoning datasets, showing reductions in tokens generated with minimal accuracy loss. The training procedure is highlighted as computationally efficient, requiring few RL steps. The paper also discusses theoretical guarantees, ablation studies on advantage normalization, and an identified bias in the RLOO implementation that can lead to length reduction even without an explicit penalty.

**Questions:**

Please refer to the weaknesses.

**Ethical Concerns:**

["NO or VERY MINOR ethics concerns only"]

**Final Justification:**

Thanks for the authors rebuttal, which addresses my most concerns. I keep my positive score.

**Limitations:**

yes

**Paper Formatting Concerns:**

N.A.

**Quality:**

3

**Strengths And Weaknesses:**

[Strengths]
The paper tackles the high inference costs associated with large reasoning LLMs, which can be crucial.

The method successfully achieves reductions in token usage while preserving accuracy.

The introduction of the `α` hyperparameter provides a way to control the token-accuracy trade-off.

[Weaknesses]
1. The theoretical analysis in Sec. 4.2, which claims accuracy is preserved, is based on simplified assumptions. This contrasts with the empirical results, where accuracy consistently drops with increasing efficiency. The paper would be more convincing if it included an analysis of this theory-practice gap.
2. The authors hypothesize that the 7B model's smoother performance curve is due to its lower learning rate but do not substantiate this claim. It is worth clarifying if sufficient compute resources are available for the 1.5B model. If so, aligning its learning rate with the 7B model's could provide direct evidence for this assertion.
3. Ideally, the hyperparameter α should be the primary factor controlling the accuracy-efficiency trade-off. However, it’s unclear whether other parameters, such as the learning rate, affect the shape and steepness of this curve. If the trade-off's characteristics are also highly sensitive to other parameters, the method's claimed simplicity would be weakened.
4. The analysis of what is being shortened is insufficient. While the example of calculating 1+1 suggests the removal of conversational filler, it remains unclear if, for complex problems, the model learns a more concise linguistic style or a fundamentally more efficient reasoning process (e.g., fewer logical steps). A qualitative analysis on more challenging tasks is needed to clarify the nature of the efficiency gains.
5. The experiments do not sufficiently decouple the benefits of the online RL approach from the specific reward design. By comparing primarily against offline baselines (SFT, DPO) , it is unclear if the performance gain stems from the online nature of the algorithm itself rather than the proposed reward function. A more direct comparison against a naive online RL baseline would be necessary to isolate this variable.

---

> ### Author Rebuttal · Authors · 2025-07-31
>
> We thank the reviewer for their thorough review. We address their main concerns below:
>
> 1. **Theory-practice gap**
>
> We thank the reviewer for highlighting this important observation. Indeed, there exists a gap between theory and empirical results, which arises because our theoretical analysis in Section 4.2 is intentionally simplified. It primarily serves as a sanity check, demonstrating the method’s soundness under idealized assumptions without confounding factors like model capacity and training dynamics. We agree that understanding and explicitly addressing this theory-practice gap is valuable. In the revision, we will include a more detailed discussion to clearly outline and analyze how practical factors, such as model capacity, training dynamics, and optimization nuances, contribute to the observed discrepancies between theoretical predictions and empirical outcomes.
>
> 2. **Learning rate experiment for the 1.5B model**
>
> We agree with the reviewer that this ablation would be highly informative. However, despite its simplicity, we were unable to conduct it within the limited timeframe of the rebuttal phase due to resource constraints on our shared infrastructure. Note that each 1.5B model training takes about 80 GPU hours. Doing this for 3 seeds and 5 values of $\alpha$ would be around 1200 GPU hours. We appreciate the suggestion and will make a strong effort to include this experiment in the final version of the manuscript.
>
> 3. **Steepness of the tradeoff curve and its relation to other hyperparameters**
>
> We thank the reviewer for the observation. Factors such as the batch size, training dataset, learning rate, and others generally affect the learning dynamics of a reinforcement learning training setup. We expect that similar consideration to apply in this case: although $\alpha$ is the primary control knob, other hyperparameters may affect the learning dynamics. Unfortunately we are unable to conduct meaningful hyperparameter sweeps due to the short rebuttal time and the lab’s finite compute resources.
>
> 4. **What is being shortened?**
>
> Thanks for pointing this out, we will add a section on how the model learns to shorten its CoT.
>
> To analyze this question, we look at both **syntactic and semantic** differences in the chain-of-thought before and after our training procedure.
>
> For semantic differences, we note that reasoning models have certain “macro”-behaviours such as backtracking, verification and exploration. These are generally triggered by certain keywords such as “wait”, “actually”, “on second thought” for backtracking, “alternatively”, “another way” for exploration and “check”, “verify”, “confirm” for verification.
>
> We compute the average number of times the 7B model performs such behaviours on the MATH500 test set and report the results in the Table below.
>
> | Method     | # Verifications | # Backtracking | # Exploration |
> |------------|----------------------|---------------------|--------------------|
> | R1         | 4.596                | 19.142              | 6.33               |
> | $\alpha=0.05$  | 3.288                | 8.894               | 6.99               |
> | $\alpha=0.1$   | 3.062                | 9.906               | 5.322              |
> | $\alpha=0.2$   | 1.53                 | 5.562               | 1.432              |
> | $\alpha=0.4$   | 1.176                | 6.388               | 2.13               |
>
>
> These numbers explain the changes in the model behaviour at a macro-level. We observe that the number of times the model verifies, backtracks and explores generally goes down as we increase the strength of compression using the $\alpha$ parameter.
>
> On a syntactic level, we observe that increasing the strength of regularization using $\alpha$ leads to shorter chain-of-thoughts where the model is generally less verbose. For instance, here is an example CoT for a prompt for the $\alpha=0.2$ and $\alpha=0.4$ models:
>
> **Prompt:**
>
> ```
> Kathy needs money for her trip to Europe. If she has 300 US dollars in the bank but wants to withdraw half of it in British pounds and half of it in euros, how many more euros than pounds will she have? Assume 1 pound = 1.64 USD and 1 euro = 1.32 USD, and round to the nearest whole number.
> ```
>
> **Response ($\alpha=0.2$)**
>
> ```
> <think>
> First, Kathy has $300 USD. She wants to withdraw half in British pounds and half in euros, so each amount will be $150.
>
> To convert $150 to pounds, I'll divide 150 by the exchange rate of 1.64. This gives approximately 91.46 pounds.
>
> For the euros, I'll divide $150 by the euro exchange rate of 1.32, resulting in approximately 113.64 euros.
>
> Finally, subtracting the pounds from the euros gives about 22.18, which rounds to 22 euros.
> </think>
>
> Kathy has \$300 USD. She wants to withdraw half in pounds and half in euros:
>
> \[
> \text{Amount in pounds} = \frac{150}{1.64} \approx 91.46 \, \text{GBP}
> \]
> \[
> \text{Amount in euros} = \frac{150}{1.32} \approx 113.64 \, \text{EUR}
> \]
>
> The difference is:
>
> \[
> 113.64 - 91.46 = 22.18 \approx 22
> \]
>
> Kathy will have \(\boxed{22}\) more euros than pounds.
> ```
>
> **Response ($\alpha=0.4$):**
>
> ```
> <think>
> First, calculate the amounts in USD to be converted to pounds and euros.
>
> Next, convert each amount to the respective currency.
>
> Finally, find the difference between the euro and pound amounts.
> </think>
>
> Kathy withdraws \$150 in pounds and \$150 in euros.
>
> Pounds: 150 / 1.64 ≈ 91.46
>
> Euros: 150 / 1.32 ≈ 113.64
>
> Difference: 113.64 - 91.46 ≈ 22.18
>
> Rounded to the nearest whole number: \boxed{22}
> ```
>
> Qualitatively, these responses are the same but the response for $\alpha=0.2$ is more verbose than the response for $\alpha=0.4$ due to unnecessary LaTeX tokens and “filler” tokens (for instance, compare “This gives approximately 91.46 pounds.“ with “ ≈ 91.46”) which don’t contribute semantically and could be removed.
>
>
> 5. **Naive online RL baseline**
>
> We are unsure what the reviewer means by online baseline. If the reviewer means running the method with $\alpha$ is set to 0 (e.g, policy gradient with no length control), this experiment is part of the result that we present. When $\alpha$ is 0, the reward function is just correctness and is not affected by response length.

---

> > ### Comment · Reviewer_YQoo · 2025-08-06
> >
> > Thanks for the authors rebuttal, which addresses my most concerns. I keep my positive score.

---

### Official Review · Reviewer_kBhE · 2025-06-30

**Clarity:** 4
**Significance:** 2
**Originality:** 2
**Rating:** 4
**Confidence:** 4

**Summary:**

This paper adds a new regularizer term in the reward computation of language models to train them to produce more concise reasonings. The results show that the model can vary its length of the reasoning by varying the amount of the regularization with minimal impact to the final accuracy.

**Questions:**

N/A

**Ethical Concerns:**

["NO or VERY MINOR ethics concerns only"]

**Final Justification:**

I retain my final score of weak accept. I think the paper has merit of being a simple method for compressing reasoning length. However, it has limited significance as the conciseness comes at a cost of the reasoning performance.

**Limitations:**

While the theme of the paper seems to deal with reasoning, the conciseness reward is very general. It would be good to see it used in other settings than mathematical reasoning too.

**Quality:**

3

**Strengths And Weaknesses:**

Strengths
- The proposed method is simple and effective.
- The experiments seem thorough on 3 different datasets (MATH500, GSM8K and AIME 2024).
- The results show a clear interpolation between conciseness and performance.
- The summary of concurrent work is helpful given the vast amount of LLM literature today.
- The baseline design seems sufficient.

Weaknesses
- Impact: There is currently still a tradeoff between the length and performance. So it seems that conciseness does not help reasoning in general but functions more like a constraint. This makes the work less influential unless the user has a budget constraint on the number of tokens. Moreover, there is a burden to train different reasoning models with different alpha values.
- Significance: Although the proposed method is simple and effective, and the study is thorough, the method may seem too simplistic for a conference contribution as there is simply an addition of reward based on the length of the generation, and everything is well under expectation: with stronger penalty we trade off some performance.

Minor:
- Figure 4 needs axis titles.
- It would be great to see more qualitative examples (App. B only provides one).

---

> ### Author Rebuttal · Authors · 2025-07-31
>
> We thank the reviewer for their thoughtful review. We’re glad that they found our technique simple and effective. We address their main concerns below:
>
> 1. **Questions on impact**
>
> *“There is currently still a tradeoff between the length and performance. This makes the work less influential unless the user has a budget constraint on the number of tokens. Moreover, there is a burden to train different reasoning models with different alpha values.”*
>
> We appreciate the reviewer pointing out the existing tradeoff. Certain practical scenarios, such as deploying LLMs on resource-constrained devices (e.g., mobile platforms), impose strict limitations on memory and latency. Therefore, it becomes crucial to explore approaches that enable reasoning models to operate effectively under these conditions.
>
> Similar trade-offs already exist in related methods—for instance, knowledge distillation creates smaller, computationally efficient models at the expense of some accuracy. Our approach specifically targets reasoning models, aiming to reduce inference-time computation by encouraging shorter yet effective reasoning chains. This directly addresses deployment concerns like inference cost reduction and it improves user experience through lower latency.
>
> We appreciate the reviewer highlighting his concern regarding the burden to train different models with different values of $\alpha$. It's worth noting that post-training models to reason efficiently requires relatively limited computational resources, typically around 100 gradient steps, compared to the extensive resources needed to initially train large reasoning models. Specifically, our approach uses at least two orders of magnitude less computational effort compared to the original training phase (for reference, see Section 3.2 in [1]).
>
> Moreover, one could potentially generate a **family of reasoning models** exhibiting varying levels of token usage and efficiency, for instance, OpenAI’s offerings with different ‘tiers’ of thinking effort such as o4-mini and o4-mini-high.
>
> 2. **Questions on significance**
>
>
> We acknowledge that our method is simple, but we view this as strength as it makes the method broadly applicable. As we are entering an era where models think longer and longer (e.g, multiple hours for the recent IMO results [2]), being efficient is an increasingly urgent need for a system’s success and adoption. We believe that our method serves as a strong starting point towards these goals.
>
>
> 3. **Minor details**
>
> Thanks for reading our paper so carefully, we will definitely improve on these points in the final version of the manuscript.
>
> 4. **Non-mathematical reasoning**
>
> We are unsure of what the reviewer means by other settings. Is it for the training dataset or evaluation datasets?
>
> If it is for the evaluation datasets, we wish to point out that we already evaluate non-mathematical benchmarks such as Logical Deduction and CommonSenseQA to assess the method’s generalization and we see similar trends as we see on the mathematical benchmarks. Extended results for these tasks are present in Section C of the Appendix.
>
> If the reviewer meant, how would you train such a model without mathematical reasoning datasets: this is a great point. Extending this training procedure to domains without binary 0/1 rewards is challenging and is a question for further research. We will add this to the Limitations of the paper.
>
> **References:**
>
> [1] ProRL: Prolonged Reinforcement Learning Expands Reasoning Boundaries in Large Language Models, arXiv 2025
>
> [2] Luong and Lockhart, Advanced version of Gemini with Deep Think officially achieves gold-medal standard at the International Mathematical Olympiad

---

> > ### Comment · Reviewer_kBhE · 2025-08-06
> >
> > I thank the authors for their response. I remain on the positive side as I think the paper is simple method for compressing reasoning length. However, conciseness does not produce a stronger reasoning model, which is still a weakness in my opinion.

---

### Official Review · Reviewer_ruBd · 2025-07-03

**Clarity:** 3
**Significance:** 3
**Originality:** 2
**Rating:** 4
**Confidence:** 3

**Summary:**

This paper introduces a novel  method that  reduces the inference cost for reasoning models while minimally affecting its accuracy. This approach  focuses on reducing the deployment cost by reducing the inference cost of the same model by minimizing token usage. Experimental evaluation verify its effectiveness.

**Questions:**

1. How does this method address the need to perform multiple full rollouts per update, followed by PPO optimization (as illustrated in Figure 2 and described in Section 4.1), which leads to significant GPU/TPU time and memory demands?

2. At Equation (11), the authors assume that $f$ is fixed and constant, yet they fail to account for its implicit dependence on $\theta$ (as defined in Eq. (5)), causing the objective to change with the policy even though it’s treated as a constant when deriving the optimum and making the argument self-contradictory.

3. This assumption 4.1 requires that there exists some $\theta$ capable of exactly realizing any arbitrary distribution. This assumption is strong. Could you please provide appropriate references

**Ethical Concerns:**

["NO or VERY MINOR ethics concerns only"]

**Final Justification:**

Although the concern on interpretability still exists, I would raise the rating from 3 to 4.

**Limitations:**

yes

**Paper Formatting Concerns:**

There is no major formatting issues

**Quality:**

2

**Strengths And Weaknesses:**

Strengths:
1. This paper is well-written and easy to follow
2. This method is flexible by integrating with standard policy‐gradient fine‐tuning without altering model architecture
3. The empirical evaluations are sufficient


Weaknesses:
1. The abstract is not concise enough
2. I have some reservations about the motivation. It primarily encourages the model to produce correct answers with short chains-of-thought. While this can effectively improve inference efficiency (particularly speed), it inevitably comes at the cost of accuracy. From my perspective, longer reasoning chains also bring benefits during training, as they promote better interpretability and richer contextual understanding.
3. This method has high demands for GPU/TPU time and memory
4. The assumptions in theoretical assessments are strong and some theoretical results remain unclear.

---

> ### Author Rebuttal · Authors · 2025-07-31
>
> We thank the reviewer for their comments. We appreciate the fact that they thought our paper was well written and that our evaluations were sufficient and the algorithm flexible.
>
> Below, we address the reviewer’s major concerns:
>
> 1. **Abstract is not concise enough**
>
> We thank the reviewer for pointing this out. We agree that the abstract can be made more concise. We will address this in our final version of the manuscript.
>
> 2. **Reservations about the motivation**
>
> *“I have some reservations about the motivation. It primarily encourages the model to produce correct answers with short chains-of-thought. While this can effectively improve inference efficiency (particularly speed), it inevitably comes at the cost of accuracy. From my perspective, longer reasoning chains also bring benefits during training, as they promote better interpretability and richer contextual understanding.”*
>
> Thank you for this valuable feedback. We agree that longer reasoning chains can indeed enhance interpretability and contextual understanding during training. However, certain practical scenarios, such as deploying LLMs on resource-constrained devices (e.g., mobile platforms), impose strict limitations on memory and latency. Therefore, it becomes crucial to explore approaches that enable reasoning models to operate effectively under these conditions.
> Similar trade-offs already exist in related methods—for instance, knowledge distillation creates smaller, computationally efficient models at the expense of some accuracy. Our approach specifically targets reasoning models, aiming to reduce inference-time computation by encouraging shorter yet effective reasoning chains. This directly addresses deployment concerns like inference cost reduction and it improves user experience through lower latency.
> We appreciate the reviewer's suggestion and will expand the motivation section of our paper to thoroughly clarify the balance between efficiency gains and the potential accuracy trade-offs, along with concrete use-case examples.
>
>
> 3. **High compute requirements**
>
> We appreciate the reviewer's concern and acknowledge that our method, employing reinforcement learning (RL), can demand substantial GPU/TPU resources. We have explicitly mentioned this limitation in the paper [Line 322]. However, alternative methods addressing the same problem, such as supervised fine-tuning (SFT) or Direct Preference Optimization (DPO), do not achieve performance comparable to online RL, as demonstrated in Figure 1. This outcome aligns with the established understanding that on-policy, online RL tailored to the specific objective generally yields superior performance compared to offline methods.
>
> Moreover, it's worth noting that post-training models to reason efficiently requires relatively limited computational resources, typically around 100 gradient steps, compared to the extensive resources needed to initially train large reasoning models. Specifically, our approach uses at least two orders of magnitude less computational effort compared to the original training phase (for reference, see Section 3.2 in [1]).
>
>
> 4. **Questions regarding theoretical assumptions**
>
> We appreciate the reviewer's valuable observation regarding the strength of our theoretical assumptions and the clarity of certain theoretical results. Indeed, our theoretical framework is intentionally simplified, focusing not on generality but rather on clearly illustrating the fundamental mechanisms and intuition behind our algorithm in an accessible manner. We discuss some of the concerns below. Nonetheless, we recognize the reviewer's feedback as an opportunity to enhance our presentation. Accordingly, we will clarify the theoretical sections in line with the reviewer’s suggestions.
>
>
>
> &nbsp;&nbsp;&nbsp;
>
> Below, we address the reviewers detailed questions:
>
> 1. *“How does this method address the need to perform multiple full rollouts per update, followed by PPO optimization (as illustrated in Figure 2 and described in Section 4.1), which leads to significant GPU/TPU time and memory demands?"*
>
> We thank the reviewer for asking the question. Our method builds on top of standard RL libraries, e.g., OpenRLHF, VeRL, and it involves only a minor change to the reward function. In this way, we can inherit all advanced system-level optimization of existing RL codebases.
>
> 2. *“the authors assume that $f$ is fixed and constant, yet they fail to account for its implicit dependence on $\theta$“*
>
> We thank the reviewer for highlighting this point. We’ve also mentioned this in Lines 216-218 in Section 4.2. The discrepancy arises as we aim to present the core result in the simplest possible setting; as the reviewer noticed, the result strictly holds for any $f$ which is a pure function of the length independent of $\theta$.
>
> 3. *“assumption 4.1 requires that there exists some capable of exactly realizing any arbitrary distribution.”*
>
> Thanks for mentioning this detail. We will add references for this assumption.
>
> The assumption that a transformer model can represent any probability distribution over a finite set of strings can be grounded using 3 simple facts:
>
> a. Every distribution over a finite set of strings $S$ can be converted into a Probabilistic Finite State Automata (PFSA)
>
> Intuitive explanation:
>
> Assume the vocabulary is called $\Sigma$. For every prefix string $p$ in the set $S$, compute conditional probabilities for each token $t  \in \Sigma$ by calculating
>
> $Prob(t | p) = \frac{\sum Prob(\text{strings starting with prefix pt})}{\sum Prob(\text{string starting with prefix p})}$
>
> Once we construct the conditional probabilities table for all possible pairs of prefixes and tokens, we can convert this into a Probabilistic Finite State Automata.
>
> b. Every PFSA is a Weighted Finite Automata (WFA): This follows from the definition of a WFA.
>
> c. Transformers can simulate WFAs: This result has been shown in [2] by Rizvi et al.
>
>
>
> **References:**
>
> [1] ProRL: Prolonged Reinforcement Learning Expands Reasoning Boundaries in Large Language Models, arXiv 2025
>
> [2] Simulating Weighted Automata over Sequences and Trees with Transformers, AISTATS 2024

---

> > ### Comment · Reviewer_ruBd · 2025-08-05
> >
> > Thanks for your effort and detailed response. I keep my original score because of several reasons:
> > i) This rebuttal  claims that shorter CoT chains maintain performance, but do not provide any interpretability metric (e.g., faithfulness, consistency)  breakdown by reasoning length. Once the interpretability does not exist, the CoT would be meaningless.
> > ii) So, do shorter CoTs degrade answer quality for complex reasoning benchmarks?
> > iii) Can the authors clarify under what assumptions on model size, depth, or context length the Transformer simulates arbitrary WFAs? Also, could the authors quantify the approximation error?

---

> > > ### Author Response · Authors · 2025-08-07
> > >
> > > Dear Reviewer,
> > >
> > > 1. ”This rebuttal claims that shorter CoT chains maintain performance, but do not provide any interpretability metric (e.g., faithfulness, consistency) breakdown by reasoning length. Once the interpretability does not exist, the CoT would be meaningless.”
> > >
> > > Thanks for your response. Evaluating interpretability (e.g faithfulness/consistency) of CoTs is an open problem. We take the approach inspired by Chen et al. [1] and Chua et al. [2] where they add "hints" such as "A Stanford professor believes that the answer is D" for a task like MMLU [3] which involves selecting the right answer from multiple choices. Intuitively, faithfulness is measured by: “if the model changes the answer when given the hint, did it actually use the hint explicitly in its CoT?”. A faithful model would acknowledge the hint explicitly whereas an unfaithful model would still change the answer but not acknowledge the effect of the hint in its CoT.
> > >
> > > More formally, assume that the unhinted prompt is $x_u$ and the hinted prompt is $x_h$, the corresponding CoTs are $c_u$ and $c_h$, and the corresponding responses are $a_u$ and $a_h$. Then we measure the following:
> > >
> > >  $\text{Faithfulness Score} = \mathbb{E}[\mathbb{1}[c_h \text{ verbalizes the hint } h] \mid a_u \ne h, a_h = h]$
> > >
> > > To check if the model verbalized the hint, we use simple string match for terms such as "Stanford" or "Professor". We evaluate 7B models on a subset of 5k prompts from MMLU and compute their faithfulness score. The results are as follows:
> > > |Model|Faithfulness Score|
> > > |-|-|
> > > |DeepSeek-R1 Distilled|0.622|
> > > |$\alpha=0.05$|0.583|
> > > |$\alpha=0.1$|0.480|
> > > |$\alpha=0.2$|0.518|
> > > |$\alpha=0.4$|0.515|
> > > |Instruct|0.301|
> > >
> > > Our results align with previous findings [1,2] which report that instruction tuned models are generally less faithful (0.622 v. 0.301) compared to reasoning models. Our results with CoT compression also generally indicate that with more compression, we get less faithful models, but the models we get are significantly more faithful compared to the non-reasoning models (0.480 v. 0.301). Altogether our findings align with those in [2], which also reports that using length penalties could result in less faithful models.
> > >
> > > We would be happy to include this study in the camera ready, should the manuscript be accepted.
> > >
> > > 2. ‘do shorter CoTs degrade answer quality for complex reasoning benchmarks?’
> > >
> > > We would like to reiterate the discussion with reviewer YQoo where we discuss what is exactly being shortened in the CoT.
> > >
> > > To analyze this question, we look at both syntactic and semantic differences in the CoT before and after our training procedure. For semantic differences, we note that reasoning models have certain “macro”-behaviours such as backtracking, verification and exploration. These are generally triggered by certain keywords such as “wait”, “actually”, “on second thought” for backtracking, “alternatively”, “another way” for exploration and “check”, “verify”, “confirm” for verification.  We compute the average number of times the 7B model performs such behaviors on the MATH500 test set and report the results in the Table below.
> > >
> > > |Method|#Verifications|#Backtracking|#Exploration|
> > > |-|-|-|-|
> > > |R1|4.596|19.142|6.33|
> > > |$\alpha=0.05$|3.288|8.894|6.99|
> > > |$\alpha=0.1$|3.062|9.906|5.322|
> > > |$\alpha=0.2$|1.53|5.562|1.432|
> > > |$\alpha=0.4$|1.176|6.388|2.13|
> > >
> > > These numbers explain the changes in the model behaviour at a macro-level. We observe that the number of times the model verifies, backtracks and explores generally goes down as we increase the strength of compression using the parameter.
> > > On a syntactic level, we observe that increasing the strength of regularization using  leads to shorter chain-of-thoughts where the model is generally less verbose. For an example of this, please refer to the discussion with Reviewer YQoo. We omit the example here because of a lack of space.
> > >
> > > 3. Can the authors clarify under what assumptions on model size, depth, or context length the Transformer simulates arbitrary WFAs? Also, could the authors quantify the approximation error?
> > >
> > > For discussions on the model size and depth required to represent WFAs, we would like to point the reviewer to Theorem 2 in [4]:
> > >
> > > Transformers can approximately simulate all WFAs with $n$ states at length $T$, up to arbitrary precision $\epsilon>0$, with depth $O(\log T)$, embedding dimension $O(n^2)$, attention width $O(n^2)$, MLP width $O(n^4)$ and $O(1)$ attention heads. They also mention that the size of the construction does not depend on the approximation error $\epsilon$ allowing them to achieve arbitrary precision without compromising the size of the model.
> > >
> > > **References:**
> > >
> > > 1. Reasoning Models Don’t Always Say What They Think, arXiv 2025
> > > 2. Are Deepseek R1 and Other Reasoning Models more Faithful?, arXiv 2025
> > > 3. Measuring Massive Multitask Language Understanding, ICLR 2021
> > > 4. Simulating Weighted Automata over Sequences and Trees with Transformers, AISTATS 2024

---

> > > > ### Comment · Reviewer_ruBd · 2025-08-08
> > > >
> > > > Thanks for your detailed response and sufficient empirical evaluation. Although the concern on interpretability still exists, I would raise the rating from 3 to 4. The authors are encouraged to add more discussion about this issue in revised manuscript.

---

### Decision · Program_Chairs · 2025-09-17

**Decision:**

Accept (poster)

**Comment:**

# Summary

The paper proposes a RL based post-training procedure that samples multiple responses and provides highest reward for shortest correct answers and lowest for incorrect answers. As the proposed reward function is a combination of accuracy and response length, the response length term can be seen as a regularizer. The weight of the regularizer can be varied to control the verbosity of the reasoning model.

The authors demonstrate their approach with DeepSeek-R1-Distill-Qwen 1.5B and 7B model on three math and reasoning datasets (MATH500, GSM8K, AIME), showing reductions in tokens generated with minimal accuracy loss. The proposed training objective is also theoretically justified to preserve accuracy and achieves the token reduction.

# Strengths

- **Simplicity and Practicality**: The proposed approach is simple, requires small modification to the reward function and is theoretically justified. The problem is of interest given the high cost of these models and the as the regularizer is controlled by single parameter it is easy to manipulate.
- **Empirical Evidence**: Empirical Evidence suggests the approach works in practice. The paper also highlights the bias in existing RLOO approach that favors shorter reasoning text and compares against it.
- **Strong Rebuttal**: The authors provided quantified responses  to almost all of the  extremely concerns raised by the reviewers. The empirical evidence addressed all most all of the concerns.


# Weakness

- **Baseline**: The original draft does not compare the proposed approach against any efficient reasoning approaches from the literature. During rebuttal, they cite lack of open-source code for this and include O1-pruner in their comparison for some dataset. But it is not clear which dataset and/or how big it is.



# Unaddressed Minor Concerns

The various minor concerns which I list below, but these do not really take away the significance of the contribution.
- **Consistency of reasoning tokens**: The shorter reasoning text might be less interpretable, consistent or less faithful. The authors do not have any analysis about it in the original draft. However, they present some initial evidence during rebuttal that suggests all is not lost. (Reviewer ruBd)
- **Generalization to other LLMs**:  It is not clear if the proposed approach only works for reasoning models (that are specifically trained to generate reasoning tokens) or it would also work for say pretrained models or smaller, less powerful models. Authors do provide initial observations by using their approach on Qwen2.5-3b base model, but comprehensive evaluation is missing.  (Reviewer vDnZ)


# Recommendation Justification

The paper proposed a very practical approach for efficient reasoning by modifying the reward function used for training. All major concerns from the reviewers were addressed by the authors during rebuttal with additional experiments. Specifically, the authors added following experiments during the rebuttal which makes the paper much stronger.

We are in the discussion phase now. With the average rating of 4.0 the paper is almost at the border. So I want to highlight some of the concerns raised during the rebuttal and confirm if these are addressed or warranty discussion.
- Analysis of what is being shortened.
- Comparison to  O1-pruner, an existing approach for efficient-reasoning.
- Comparison of proposed approach on base-model instead of the finetuned model.
- Evaluation of faithfulness of the model.

Given the consistent reviewer support (average rating 4.0, all 4 "borderline accept"), the thorough rebuttal, and the relevance to the community, I believe the paper merits inclusion in the conference.